# Genetic architecture and mechanisms of host-microbiome interactions from a multi-cohort analysis of outbred laboratory rats

Hélène Tonnelé [1,2,24], Denghui Chen [3,24], Felipe Morillo[1,2], Jorge Garcia-Calleja [4], Apurva S. Chitre[3], Benjamin B. Johnson[3], Thiago Missfeldt Sanches[3], Riyan Cheng[3], Marc Jan Bonder [5], Antonio Gonzalez[6], Tomasz Kosciolek[6], Anthony M. George[7], Wenyan Han [8], Katie Holl [9], Aidan Horvath[10], Keita Ishiwari[7,11], Christopher P. King[12], Alexander C. Lamparelli[12], Connor D. Martin[7,11], Angel Garcia Martinez [8], Alesa H. Netzley[10], Jordan A. Tripi[12], Tengfei Wang[8], Elena Bosch [4], Peter A. Doris [13], Oliver Stegle [14,15], Hao Chen [8], Shelly B. Flagel[16,17], Paul J. Meyer [12], Jerry B. Richards[7,11], Terry E. Robinson[10], Leah C. Solberg Woods [18], Oksana Polesskaya[3], Rob Knight [6,19,20,21,22], Abraham A. Palmer [3,23] ✉ & Amelie Baud [1,2] ✉

The intestinal microbiome influences health and disease. Its composition is affected by host genetics and environmental exposures. Understanding host genetic effects is critical but challenging in humans, due to the difficulty of detecting, mapping and interpreting them. To address this, we analyse host genetic effects in four cohorts of outbred laboratory rats exposed to distinct but controlled environments. We show that polygenic host genetic effects are consistent across cohort environments. We identify three replicated microbiome-associated loci, one of which involves the sialyltransferase gene *St6galnac1* and *Paraprevotella*. We find a similar association in a human cohort, between *ST6GAL1* and *Paraprevotella*, both of which have been linked with immune and infectious diseases. Moreover, we find indirect (i.e. social) genetic effects on microbiome phenotypes, which substantially increase the total genetic variance. Finally, we identify a novel mechanism whereby indirect genetic effects can contribute to "missing heritability".

The mammalian intestinal tract is home to trillions of metabolically active, immunomodulating microorganisms. In humans, the composition of the gut microbiome is correlated with many diseases and, in some cases, manipulations of the microbiome have suggested that these relationships are causal[1]. The composition of the gut microbiome is affected by a variety of factors, including maternal seeding at birth[2–5], diet[6], medication[6,7], host genetics effects[8–15] and acquisition of microbes from family members and unrelated social partners[16–18]. A better understanding of how these factors act could open new avenues for medicine.

Host genetic effects have been detected in humans[8–15], but questions remain as to the proportion of bacteria affected, how much variation in these bacteria is due to host genetic effects, whether host genetic effects are consistent across environments, whether they act by influencing the host environment (e.g., diet), and what the underlying genetic loci are. Only two microbiome-associated loci have been replicated so far: the *LCT* lactase persistence locus[8,9,19] and the *ABO* blood group locus, which interacts with the *FUT2* secretor status locus[12,13].

Furthermore, recent evidence indicates that many gut bacteria can be transmitted vertically (from parents to offspring) and/or horizontally (between non-parental individuals)[16–18], but these findings have not yet been integrated with findings on host genetic effects. In particular, whether host genetic effects in one individual can alter the microbiomes of other individuals through microbial transmission of heritable bacteria remains unexplored.

Many factors make it difficult to detect, quantify, map and interpret host genetic effects in humans. Environmental factors such as diet, medication and social contacts affect the composition of the microbiome, and few human datasets have sufficient information on these variables. Even when information is available, whether host genetic effects are mediated by heritable components of host environment is difficult to establish[10]. In addition, technical variation in the protocols and methods used to profile the microbiome complicates efforts to meta-analyse the data[11]. Given the difficulties of studying host genetic effects in humans, studies in other species are needed.

Due to the shared anatomy and physiology of mammals, laboratory rodents are a model of choice to study host-microbiome interactions[20]. Populations of laboratory rodents derived from multiple founders through many generations of outbreeding provide high levels of genetic diversity, fewer rare variants and higher levels of linkage disequilibrium (LD) compared to humans. Higher levels of LD mean that fewer independent tests are performed in genome-wide association studies (GWAS), and smaller sample sizes can be used. The levels of LD observed in some outbred laboratory rodent populations remain compatible with the identification of causal genes at associated loci[21,22]. Crucially, laboratory rodents live in a controlled environment, greatly reducing noise and confounding effects from both genetic and non-genetic factors influencing the gut microbiome.

In this study, we conducted a comprehensive analysis of host genetic effects in four cohorts of outbred "Heterogeneous Stock" (HS) rats, a population established from eight inbred strains[23] and thereafter maintained using an outbreeding scheme (Fig. 1A). The 4154 male and female rats included in this study were part of a multisite project examining the genetic basis of physiological and behavioural traits[24–27], so we had access to genotype information at about 5.5 million single nucleotide polymorphisms (SNPs)[28]. The majority of the SNPs (>98%) were common (MAF > 0.01) (Supplementary Fig. 1A) because they are derived from one or more of the eight founders (hence MAF ≥ 1/8), and LD between neighbouring variants extended slightly further than in humans (Supplementary Fig. 1B, average distance between SNPs with LD $R^2 > 0.9$ was $111 \pm 788$ Kb in HS rats, compared to $39 \pm 33$ Kb in humans).

The rats included in this microbiome study belonged to four distinct cohorts, referred to as NY, MI, TN1 and TN2, which refer to the U.S.A. states in which they were reared. The four cohorts differed in terms of rearing facilities, age and fasting state when the caecal (gut) microbiome was sampled, among other known and presumably unknown differences (Fig. 1B and Supplementary Table 1). In contrast, cross-cohort genetic differences were minimised by distributing the litters across the NY, MI and TN2 cohorts. Finally, all microbiome samples were profiled in a single laboratory using the same protocol. This experimental design was ideal to evaluate the consistency of host genetic effects across environments.

In this work, we show that polygenic host genetic effects are consistent across cohort environments. We identify three replicated microbiome-associated loci, one of which involves the sialyltransferase gene *St6galnac1* and *Paraprevotella*. We find a similar association in a human cohort between *ST6GAL1* and *Paraprevotella*, both of which have been linked with immune and infectious diseases. Moreover, we find evidence that the composition of the gut microbiome is affected by the genetic makeup of cage mates (i.e. indirect genetic effects), which substantially increases the total genetic variance of microbiome phenotypes. Finally, we identify a novel mechanism whereby indirect genetic effects can contribute to "missing heritability".

A Catalan translation of the abstract and a non-specialist summary can be found at https://doi.org/10.5281/zenodo.17828116.

## Results

### Between-environment variation in the caecal microbiome

This study focused on the rat caecal microbiome, which is a model for the human colonic microbiome since the rat caecum and the human colon are the main sites of fermentation of plant-based foods[20]. The caecal microbiome of the rats was profiled by amplicon sequencing targeting the 16S gene of gut bacteria and archaea. The microbiome phenotypes we considered were amplicon sequence variants (ASVs), which are error-corrected sequences providing the greatest taxonomic resolution possible with 16S data, as well as the higher-level bacterial taxa these ASVs were assigned to.

In all four cohorts, the *Lachnospiraceae* family was the most abundant bacterial family. Other abundant families (>10% relative abundance in at least one of the cohorts) were *Oscillospiraceae_88309*, *Muribaculaceae* and *Bacteroidaceae* (Fig. 2A). As expected, the composition of the gut microbiome was more similar among rats from the same cohort ("adonis" analysis of variance $p$-value < 0.001, Fig. 2B). The variation explained by cohort ($R^2 = 0.31$) was much greater than the variation explained by sex ($R^2 = 0.0092$). 130 out of 364 bacterial families were differentially abundant between the cohorts (Bonferroni-adjusted analysis of variance $p$-value < 0.05), including the four most abundant families. In addition, within-sample (alpha) diversity was also significantly different between cohorts (analysis of variance $p$-value $= 3.28 \times 10^{-83}$, Supplementary Fig. 2).

### Polygenic host genetic effects and their consistency across environments

As a first step, we characterised polygenic host genetic effects using variance components models. We analysed the different cohorts separately to evaluate the consistency of host genetic effects across environments. We limited our analyses to ASVs and taxa present in at least 50% of the rats in a cohort, to get precise estimates of the variance components. Thus, hereafter, a microbiome phenotype refers to a given ASV or taxon in a given cohort, and we considered 569, 589, 523 and 613 microbiome phenotypes in the NY, MI, TN1 and TN2 cohorts, respectively. In all our analyses, the compositional nature of microbiome data[29] was accounted for. Even though confounding between host genetic effects and cage (i.e., shared micro-environmental) effects was avoided by design in the NY, MI and TN2 cohorts, we accounted for any potential confounding from maternal and cage effects in all our analyses by fitting jointly random effects for host genetic effects, cage effects and maternal effects (see "Methods" section on confounding and accounting for it).

Heritability is a population-specific measure. Hence, the heritability of microbiome phenotypes is best understood when it is compared to the heritability of other phenotypes measured in the same population or compared to the magnitude of other (non-genetic) effects. The SNP heritability of microbiome phenotypes was generally lower than the SNP heritability of behavioural phenotypes relevant to substance use disorder (e.g., delay discounting, cue and context conditioning, nicotine self-administration[25,26]) and physiological phenotypes (e.g., adiposity, kidney and liver weight, glycemia[24]) measured in the same rats (Fig. 3A). ASVs tended to have higher heritability than species and genera, suggesting that host genetic effects may be heterogeneous within bacterial species (Supplementary Fig. 3). Notably, the most heritable microbiome phenotypes were not necessarily the most abundant ones (Fig. 3B and Supplementary Fig. 5). Host genetic effects were comparable in magnitude to maternal effects on the microbiome and much lower than cage effects (Fig. 3C and Supplementary Fig. 4).

We next tested whether the heritability of an ASV or a taxon in one cohort was predictive of the heritability of the same ASV or taxon in

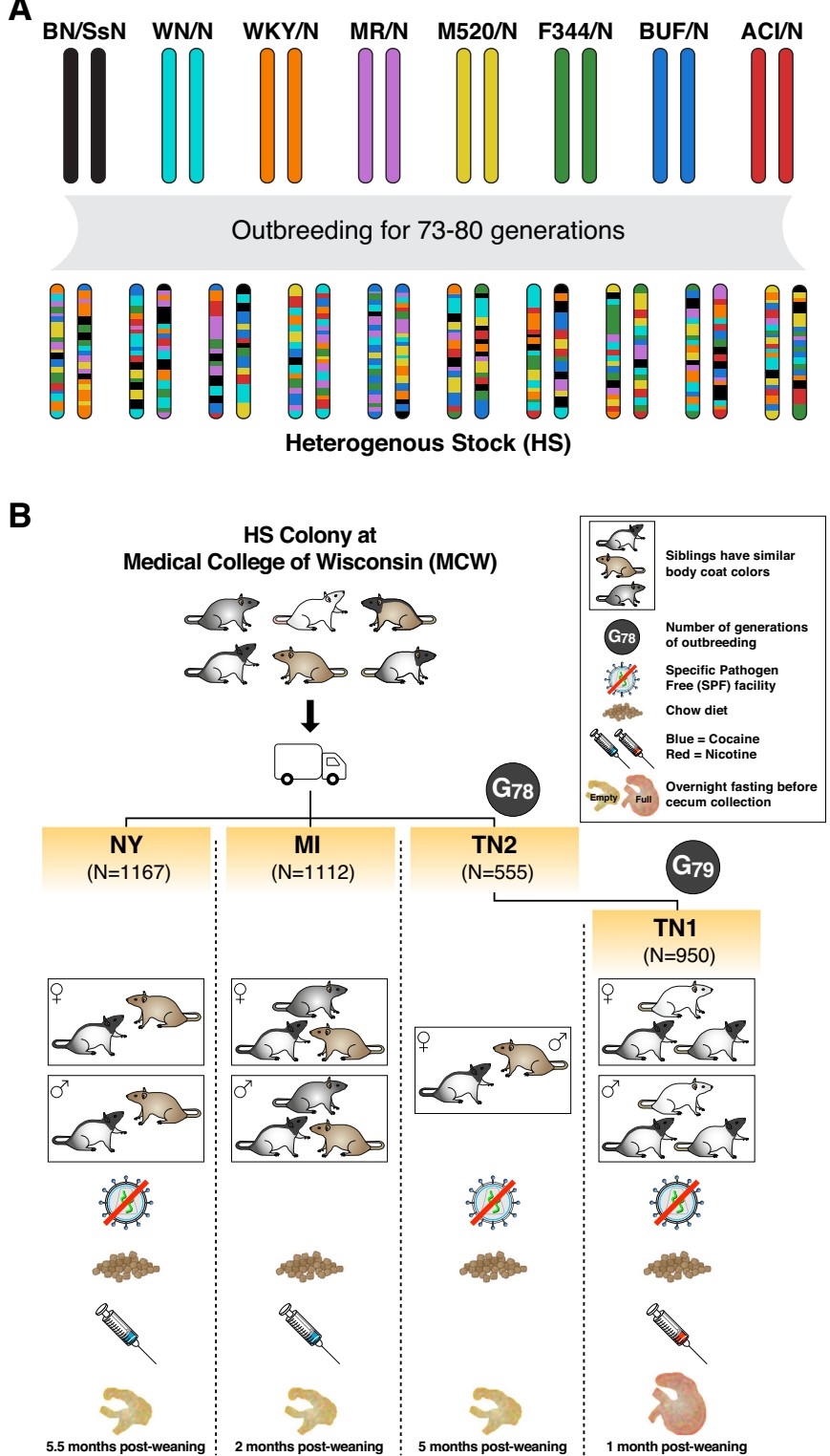

**Fig. 1 | Experimental design with Heterogeneous Stock (HS) rats. A** Provenance of the HS rats used in this study. **B** Overview of the four cohorts of HS rats included in this study. Detailed information is provided in Supplementary Table 1 and the "Methods" section.

another cohort. The heritability of ASVs and taxa measured in the largest cohort (NY) was significantly, but only modestly, correlated with the heritability of the same ASVs and taxa in two other cohorts (MI and TN2, cor = 0.31 and cor = 0.13 respectively, *p*-value < 0.05 after Bonferroni correction accounting for the six correlations tested, Fig. 3D). A low correlation (or concordance) of heritability values between two cohorts could occur, for example, if a subset of bacteria

was affected by an experimental factor and the rest was not, and the experimental factor was more variable in one cohort than in the other.

The cross-cohort concordance of heritability values does not indicate, however, whether the same molecular and biological processes underlie host genetic effects in the different cohorts ("consistent" host genetic effects). If, for a given bacteria, the same processes are at play in two cohorts, then we expect the effect sizes of

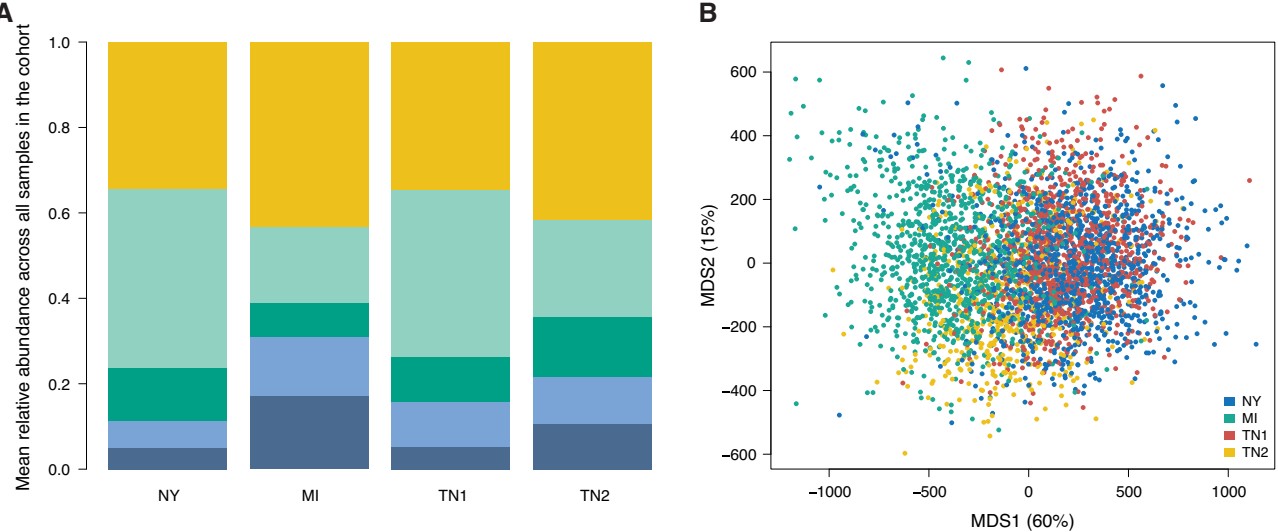

**Fig. 2 | Variation in the HS rat gut microbiome. A** Average family-level microbiome profiles in the different cohorts. From top to bottom, yellow is for the sum of abundances of lower abundance families, light green is for the *Lachnospiraceae* family, dark green is for the *Oscillospiraceae_88309* family, light blue is for the *Muribaculaceae* family, and dark blue is for the *Bacteroidaceae* family. **B** Between-sample ("beta") diversity across all four cohorts. MDSX: multi-dimensional scaling axis X.

the individual genetic variants in the two cohorts to be highly correlated (i.e., a high genetic correlation). Thus, for each ASV and taxon and each pair of cohorts, we estimated the cross-cohort genetic correlation between the ASV or taxon's abundance in one cohort and its abundance in the other cohort. We did so only for those ASVs and taxa that had a substantial heritability (>5%) in both cohorts. The cross-cohort genetic correlations were rather high (median value 0.5, median standard errors 0.3, Fig. 3E), and 60% of them were significant (FDR < 0.1), demonstrating substantial consistency of polygenic host genetic effects across cohort-associated environments. One ASV, in the genus *Anaerotruncus*, was particularly consistent across the different environments, with all between-cohort genetic correlations close to 1.

Motivated by the largely shared genetic basis of ASVs and taxa across cohorts, we analysed all samples jointly (fitting cohort as a fixed effect covariate) to identify significantly heritable ASVs and taxa. Using the joint sample (N = 3784), we found that 214 out of 546 common microbiome phenotypes (39.2%) were significantly heritable (FDR < 0.1). The correlation between abundance and heritability was not significant, but heritable microbiome phenotypes were significantly more abundant than non-heritable phenotypes (one-sided *t*-test *p*-value 0.002). Host genetic effects explained an average of 3.2% (s.d. 1.2%) of phenotypic variation for these 214 phenotypes, with a maximum heritability of 9.3% and a minimum heritability of 0.5%.

### Microbiome-associated loci and their replication across environments
We next mapped the genomic loci affecting the microbiome phenotypes measured in each cohort, considering only those ASVs and taxa present in at least 50% of the rats in each cohort. The genome-wide significance threshold (−logP ≥ 5.8), which corrects for the number of independent SNPs tested, was determined using permutations[30,31] and later adjusted to account for the number of independent microbiome phenotypes analysed in each cohort, yielding a similar cohort-wide significance threshold (−logP ≥ 8.4) in all four cohorts. 6 associations were cohort-wide significant in the NY cohort, 5 in the MI cohort, 2 in the TN1 cohort, and 0 in the TN2 cohort (Fig. 4). Seven of these cohort-wide associations were with the same locus on chromosome 1 (associated variants all in strong LD), including associations with genera *CAG-793* and *UMGS1994* detected in multiple cohorts

(Supplementary Table 2). The seven associations involved different genera in the *Firmicutes_A* phylum, providing strong evidence of pleiotropy at this locus. Another one of the cohort-wide significant associations was on chromosome 4, for the genus *CAG-485* measured in the NY cohort. Although not significant at the stringent threshold we used (−logP ≥ 8.4), associations were also detected in the MI and TN1 cohorts for this locus and this genus at the less stringent threshold for genome-wide significance (−logP ≥ 5.8, Supplementary Table 3). Finally, five of the cohort-wide significant associations were with the same locus on chromosome 10 (associated variants all in strong LD), all of them with either the genus *Paraprevotella* or an ASV in the genus *Paraprevotella*, and in three different cohorts (NY, MI and TN1, Supplementary Table 4).

Given the observed consistency of polygenic host genetic effects across cohorts, we mapped host genetic effects in the full sample (N = 3,784), fitting cohort as a fixed effect to account for differences in mean abundances between cohorts and adjusting the genome-wide significance threshold to account for the effective number of common microbiome phenotypes considered in the full sample (−logP = 8.4). This analysis did not uncover any new associations compared to the analysis of the individual cohorts (Supplementary Fig. 6). Indeed, the 11 significant associations detected with the full sample were between the three replicated loci on chromosomes 1, 4 and 10 and the previously described genera.

### Mechanisms of host genetic effects on the gut microbiome
We next adopted a systems genetics approach to gain insights into the genetic variants and genes underlying the replicated associations at the chromosome 1, 4 and 10 loci, leveraging a wealth of genomic data, including an exhaustive catalogue of SNPs and small insertions/deletions segregating in the HS population[28], PacBio HiFi long-read genome sequencing data from the HS founders and a subset of HS rats, RNA isoform sequencing data from related laboratory rat strains, and publicly available genotype and short-read genome sequencing data from wild rats[32,33].

At the replicated locus on chromosome 1 (associated with various bacteria in the *Firmicutes_A* phylum), the variants in strong LD (defined as $R^2 > 0.9$) with the lead variant formed a large LD block extending over 3.2 Mb (Supplementary Fig. 7). Such a large LD block is highly unusual in the HS population. Indeed, this was the largest LD block

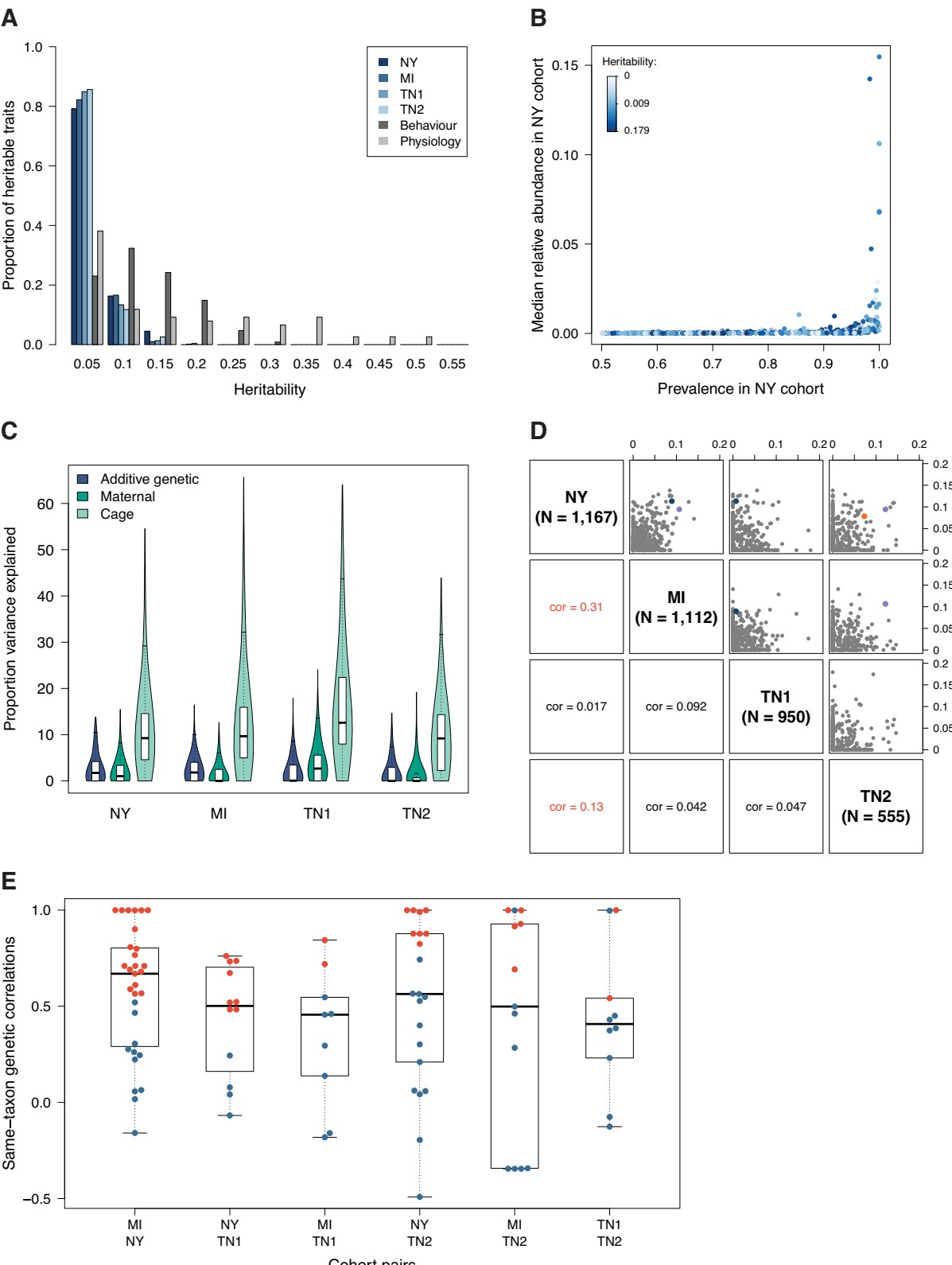

observed for any of the genome-wide significant microbiome-associated loci (median size was 0.57 Mb). Since the associations at this locus were highly significant, we looked for variants that might affect protein structure and function. Among the variants in strong LD with the lead variant, three were predicted to be probably or possibly damaging by Polyphen, all in the olfactory receptor gene *Or13a18c*, which has no obvious connection to the microbiome. On the other hand, four genes within this locus encode gel-forming mucins. In particular, *Muc2*, which encodes the most abundant mucin present in the gut and is known to play a key role in host-microbiome interactions, is contained within this locus.

At the replicated locus on chromosome 4 (associated with a single ASV in the *Muribaculaceae* family), the variants in strong LD with the lead variant also fell in a larger than usual block of LD, extending over

**Fig. 3 | Characteristics of polygenic host genetic effects. A** Comparison of the heritability of microbiome and behavioural and physiological phenotypes. **B** Relationship between prevalence, relative abundance and heritability for the microbiome phenotypes measured in the NY cohort (similar results are observed in the other cohorts, see Supplementary Fig. 5). **C** Relative magnitude of polygenic host genetic effects (SNP heritability), maternal effects and (shared micro-environmental) cage effects. **D** Pearson correlation between the heritability of one ASV/taxon in one cohort and the heritability of the same ASV/taxon in another cohort. Only ASVs and taxa that are common (prevalence >50%) in both cohorts are used. The dots corresponding to the most significantly associated ASV/taxon at each of the three replicated loci (see Fig. 4 below and Supplementary Tables 2–4) are

coloured using the same colouring scheme as in Fig. 4 (ASV_3613 in purple, ASV_18566 in blue, and ASV_5163 in orange). The dots corresponding to these ASVs/taxa were coloured only when there was a genome-wide significant association in both cohorts of the cohort pair considered (NY, MI and TN2 for ASV_3613; NY, MI and TN1 for ASV_18566; NY and TN1 for ASV_5163). **E** Genetic correlations between the abundance of an ASV/taxon in one cohort and the abundance of the same ASV/taxon in another cohort. Genetic correlations were estimated from an extension with two dependent variables of model (1) (see "Methods"). Only ASVs and taxa that are common in both cohorts and have a heritability greater than 5% were used. Correlations significantly different from 0 (LRT dof = 1 $p$-value < 0.05) are shown as red dots.

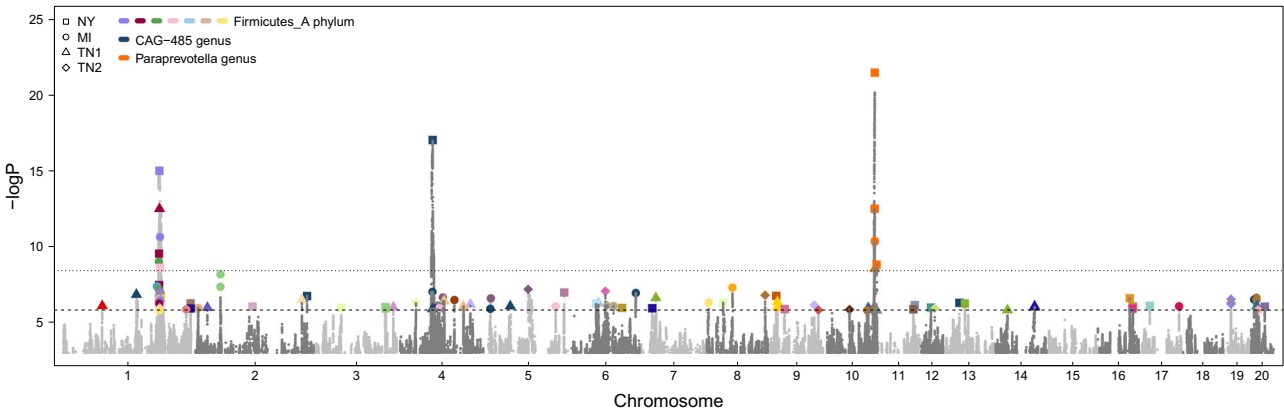

**Fig. 4 | Microbiome-associated loci.** This porcupine plot shows the association values for all microbiome phenotypes ($n = 2294$ phenotypes from four cohorts). The lower line ($-\log P = 5.8$) reflects the genome-wide significance threshold for an individual trait, which accounts for the number of independent SNPs tested; the higher line ($-\log P = 8.4$) is the adjusted significance threshold, which, in addition,

accounts for the number of independent microbiome phenotypes examined in a cohort. The larger, coloured dots highlight genome-wide significant associations for microbiome phenotypes that have an annotation at the genus level. The colour of the dot refers to the genus affected. The shape of the dot refers to the cohort in which the association was identified.

1.3 Mb (Supplementary Fig. 8). Since the associations at this locus were also very significant, we again sought to identify variants affecting protein structure and function. One deleterious variant was present in each of four genes within this locus: *Pip*, *Prss58*, *Kel* and *Tas2r139*. An effect of PIP (prolactin-induced protein) on the oral microbiome and various cultured bacterial strains was previously reported[34,35]. However, the effect of *Pip* on *Muribaculaceae*, which are mostly present in the lower gut, is novel.

The replicated locus on chromosome 10 (associated with *Paraprevotella*) harboured the most significant association of all ($-\log P = 21.5$, Fig. 5A). The variants in strong LD with the lead variant were all specific to the WN/N founder and within a small 105Kb region harbouring only three genes: *St6galnac2*, *St6galnac1* and *Mxra7*. Thirteen variants were annotated by Polyphen as possibly or probably damaging, all of them in *St6galnac1*, a gene responsible for the α2,6-sialylation of gut mucin O-linked glycans and also, potentially indirectly, N-linked glycans[36]. Increases in the depth of sequencing of a 9 kb region from the WN/N HS founder, some outbred HS rats that carried the WN/N haplotype, and wild rats suggested a triplication of *St6galnac1* (Fig. 5B and Supplementary Fig. 9). Since this region of WN/N is identical by descent to the SHRSP/A3N inbred rat strain (Supplementary Fig. 10; SHRSP is not a founder of the HS population) and a de novo assembly of the genome of SHRSP/BbbUtx is available[37], we confirmed that the *St6galnac1* triplication was present in the SHRSP/BbbUtx genome assembly (Fig. 5C). Furthermore, using IsoSeq RNA sequencing reads from lymphoid tissue of an F1 rat that was a cross between SHRSP/BbbUtx (strain with three copies of *St6galnac1*) and WKY/Utx (single copy of *St6galnac1*), we found that the two extra copies encoded a full-length transcript, contradicting the current gene

annotation of the SHRSP/BbbUtx assembly and suggesting that the copy number variant observed in the WN/N haplotype could result in increased expression of *St6galnac1*.

The causal role of *St6galnac1* was supported by an independent line of evidence: in HS rats, *Akkermansia muciniphila* and *Muribaculum intestinale* were also associated with the *St6galnac1* locus (MI cohort, not genome-wide significant, see Supplementary Table 4 and Fig. 5D–F). Since these bacteria have been shown to be strongly affected by *St6galnac1* in a mouse gene knock-out experiment by Yao et al.[36], the associations between *St6galnac1* and *A. muciniphila* and *M. intestinale* detected in HS rats further support a causal role of *St6galnac1*.

In a human cohort[9], the second most significant association for *Paraprevotella* abundance was with a locus harbouring *ST6GAL1* ($p$-value = $1.37 \times 10^{-7}$, not genome-wide significant; note that *Paraprevotella* was detected in only 451 out of 984 study participants, decreasing the power to detect a genome-wide significant association). ST6GAL1 catalyses the α2,6-sialylation of mucin N-glycans[36,38], a function shared with *St6galnac1*[36], suggesting shared modalities of host-microbiome interactions in humans and rats.

## Coupling of host genetic effects and horizontal microbial transmission

Recent studies in humans that tracked microbial strains across pairs of socially interacting individuals provided strong evidence that gut microbes can be horizontally transmitted[16–18]. However, whether some bacteria are affected by host genetic effects and are also horizontally transmitted is unknown. In addition, the quantitative impact of these joint processes on microbiome variation has not been estimated. We

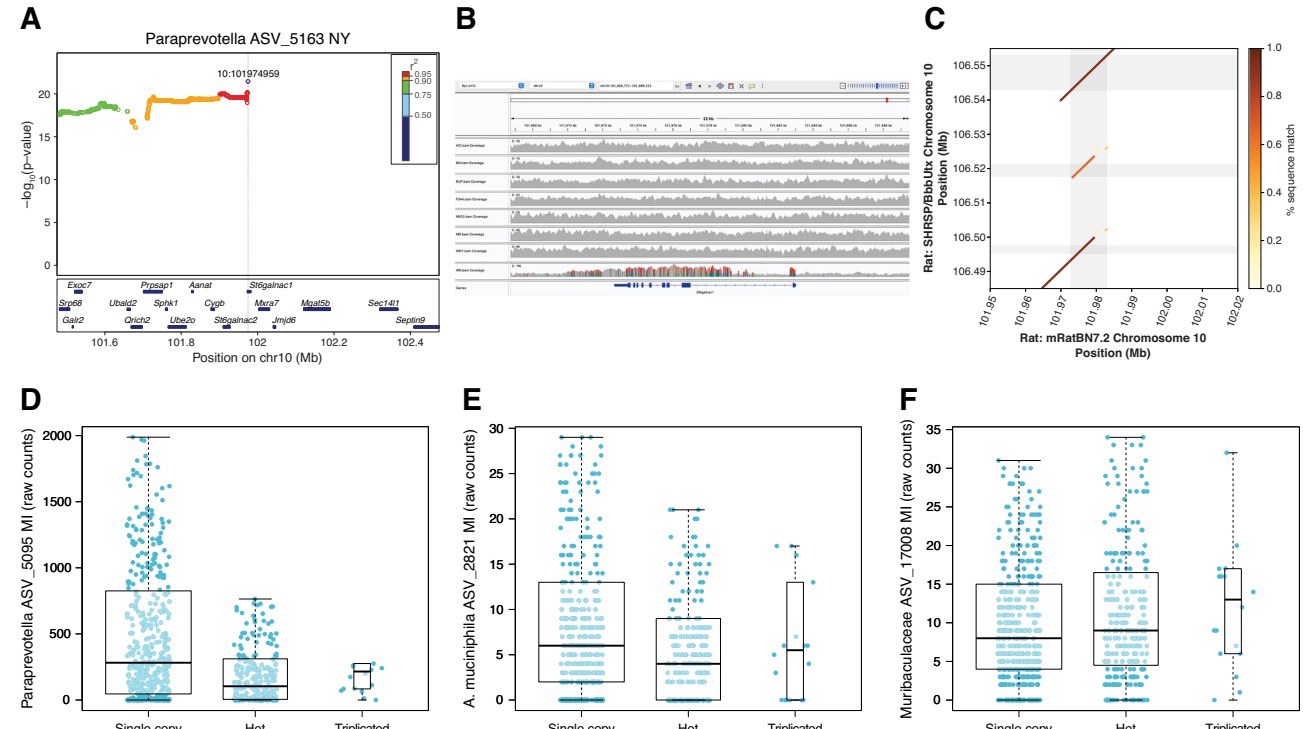

**Fig. 5 | Dissection of the mechanisms underlying the replicated association between *Paraprevotella* and the *St6galnac1* locus on chromosome 10. A** Local association (LocusZoom) plot for ASV_5163 (genus *Paraprevotella*) measured in the NY cohort and the *St6galnac1* locus. Eight genes were omitted for graphical purposes. The y-axis shows the $-\log_{10}$ p-value from an LRT with one degree of freedom comparing a model with a fixed effect for the tested SNP to a null model without (see "Methods" section on GWAS). A wider plot is shown in Supplementary Fig. 11. **B** Alignment of Illumina short reads from the eight founders of the HS population to the rat reference genome assembly (rn7). **C** Comparison of the rat reference genome assembly (rn7, from strain BN/NHsdMcwi) and the de novo genome assembly of the strain SHRSP/BbbUtx, which is identical by descent to WN/N at the locus. **D–F** Genotypic effects on *Paraprevotella*, *A. muciniphila* and *M. intestinale* ASVs that map to the *St6galnac1* locus (see Supplementary Table 4). A few outliers were omitted for better visualisation of the quartiles. Given the very small number of rats that were homozygous for the triplication, the association is driven by the difference between the other two genotype groups.

developed an approach to address these questions, based on a large body of work on "indirect genetic effects"[39–41] (IGE, genetic effects on the phenotype of an individual that arise from genotypes of *other* individuals through social interactions). This approach does not require strain-level microbiome data. As illustrated in Fig. 6A, when the abundance of a microbe is influenced by host genetic effects (hereafter "direct genetic effects" on the microbe, Mic-DGE) and that same microbe is transmitted between socially interacting individuals, IGE are expected to affect the abundance of the microbe (Mic-IGE). In that situation, Mic-DGE and Mic-IGE will arise from the same variants, hence their correlation will be close to 1.

We fitted jointly random effects for Mic-DGE, Mic-IGE, cage effects, maternal effects and other environmental effects (see "Methods" section on confounding and accounting for it). The modelling of Mic-DGE and Mic-IGE focused solely on polygenic effects. As we expected Mic-IGE to be generally weaker than Mic-DGE[42,43], we analysed all four cohorts jointly ($N = 3767$ rats) to maximise statistical power to detect Mic-IGE and to obtain precise estimates of the correlation between Mic-DGE and Mic-IGE (see "Methods"). We used a bootstrap approach to obtain the distribution of p-values expected under the null hypothesis of no Mic-IGE and verified that these p-values followed a uniform distribution, i.e., were calibrated (Supplementary Fig. 12). We then calculated a Mic-IGE p-value for each microbiome phenotype and applied a conservative Bonferroni correction to account for the total number of microbiome phenotypes considered ($-\log_{10}(0.05/546) = 4$). Three ASV phenotypes were significantly affected by Mic-IGE at this threshold. These ASVs were in different genera of the *Muribaculaceae* family, which is abundant in rodents and also found in humans[44]. At a less stringent false discovery rate (FDR)

threshold of 10%, 26 microbiome phenotypes were significantly affected by Mic-IGE. Furthermore, a comparison of observed Mic-IGE p-values against the p-values expected under the null hypothesis of no Mic-IGE suggested that many more bacteria are affected by Mic-IGE (Fig. 6B).

For the three phenotypes significantly affected by Mic-IGE (Bonferroni corrected threshold), the correlation between Mic-DGE and Mic-IGE was very close to 1 ($1.0 \pm 0.16$, $0.91 \pm 0.091$, $1.0 \pm 0.19$), providing strong evidence that horizontal microbial transmission mediates Mic-IGE. Similarly, the correlation between Mic-DGE and Mic-IGE was above 0.9 for 20 of the 26 phenotypes affected by Mic-IGE at the 10% FDR threshold.

For the same three phenotypes, we calculated the total genetic variance[45] explained by Mic-DGE, Mic-IGE and their covariance. This total genetic variance reflects the impact of host genetic effects on the individual's own microbiome phenotype and on the microbiome phenotype of its cage mates. Therefore, it captures not only the genetic variance for *acquiring* microbes but also for *transmitting* microbes[46]. The total genetic variance was 4.4, 7.7 and 4.8 times greater than the variance explained by Mic-DGE (host genetic effects) in a model without Mic-IGE (Fig. 6C), highlighting the quantitative impact of Mic-IGE on inter-individual variation in microbiome composition. The significance of this result was confirmed with permutations: when we scrambled the cage mates' assignments (see "Methods" section), the total genetic variance was much lower and never as high as the value estimated using the real (unpermuted) data (Fig. 6C). Similarly, the variance explained by Mic-IGE was much lower in the permutations (Supplementary Fig. 13A). Thus, modelling Mic-IGE revealed a substantial amount of hidden genetic variance. Modelling

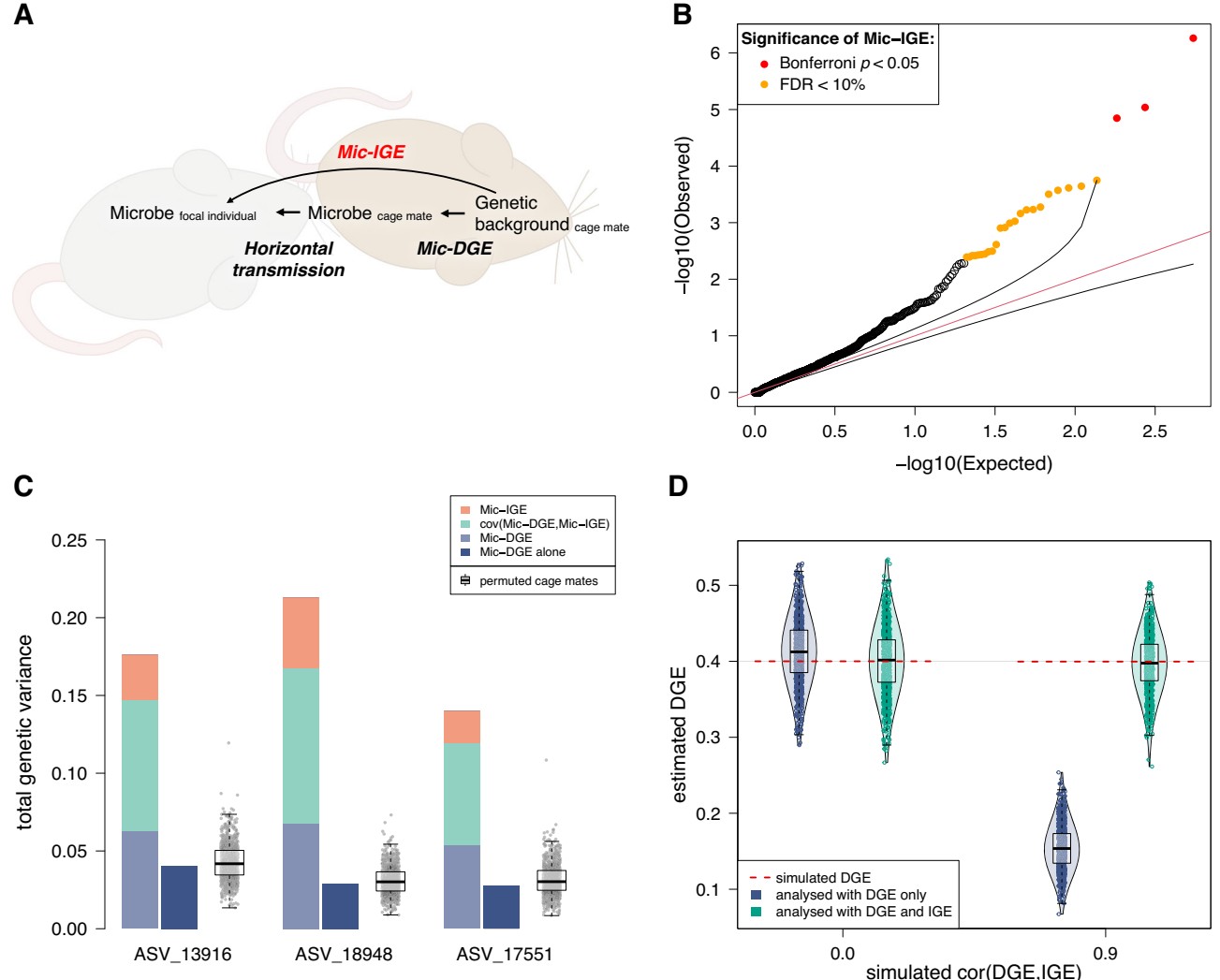

**Fig. 6 | Indirect (social) genetic effects on microbiome phenotypes (Mic-IGE). A** Schematic illustrating how Mic-IGE arise from the coupling of host genetic effects (Mic-DGE) and horizontal transmission. **B** QQ plot comparing the observed Mic-IGE $-\log_{10}$ $p$-values to the Mic-IGE $-\log_{10}$ $p$-values values expected under the null hypothesis of no Mic-IGE ($p$-values from LRT, see "Methods" section on Mic-IGE). We show in Supplementary Fig. 12 that Mic-IGE $p$-values are calibrated, i.e., follow the expected uniform distribution in the absence of Mic-IGE. **C** Bar plots comparing the total genetic variance explained by Mic-DGE, Mic-IGE and their covariance to

the genetic variance explained by Mic-DGE in a model not including Mic-IGE. The grey boxplots show the total genetic variance when cage mates' assignments are permuted (hence Mic-IGE cannot be captured). Supplementary Fig. 13A shows the variance explained by Mic-IGE when cage mates' assignments are permuted. **D** DGE (e.g., Mic-DGE) estimates from 1000 phenotypes simulated with DGE, IGE and cage effects and analysed either with a model including IGE or not. See Supplementary Fig. 14 for the MI cohort.

Mic-IGE reduced cage effects by 26%, showing that cage effects are partly due to Mic-IGE.

Finally, noticing that Mic-DGE were underestimated when Mic-IGE were not modelled (Fig. 6C), we hypothesised that, more generally, when related individuals are included in a sample, IGE occur between unrelated individuals, and DGE and IGE are positively correlated (e.g., some spousal IGE in the UK Biobank[47]), DGE might be underestimated by a model not accounting for IGE. The intuition for this hypothesis is the following: DGE make closely related individuals more phenotypically similar to each other than average; in the above-mentioned scenario, where the most closely related individuals are not the ones interacting, DGE will tend to make the individuals that interact with each other less similar to each other than average. In contrast, the positive correlation between DGE and IGE will tend to make the individuals that interact with each other more phenotypically similar than average. Thus, DGE and the positive correlation between DGE and IGE will tend to create opposite patterns of phenotypic similarity, and omitting IGE and their correlation with DGE will lead to

underestimating DGE. We tested this hypothesis using the same permutations as above and dedicated simulations (see "Methods" section) and confirmed that failing to account for IGE results in underestimating DGE (Fig. 6D, Supplementary Fig. 14 and Supplementary Fig. 13B). In our study, for the 26 microbiome phenotypes that were significantly affected by Mic-IGE at FDR < 0.1, Mic-DGE from a model not accounting for Mic-IGE were 37% smaller than Mic-DGE from a model accounting for Mic-IGE and their correlation with Mic-DGE.

## Discussion

We carried out an in-depth analysis of host genetic effects affecting the gut microbiome in four cohorts of HS rats. We demonstrated that host genetic effects are consistent across the environments experienced by the different cohorts and identified three significant microbiome-associated loci that replicated in three or four cohorts. At one of these loci, we identified the causal variant and gene, and we found a similar association in a human cohort. In addition, we showed that the coupling between host genetic effects and horizontal transmission gives

**Table 1 | Relative abundances of *A. muciniphila*, *M. intestinale* and Paraprevotella abundances in *St6galnac1* KO and WT mice, and in HS rats with either a single copy of *St6galnac1* or heterozygous for a triplication of *St6galnac1***

| | | Abundance | | | | Abundance | | |
|---|---|---|---|---|---|---|---|---|
| | | *A. muciniphila* | *M. intestinale* | | | *A. muciniphila* | *M. intestinale* | *Paraprevotella* |
| Yao et al.' Fig. 7D | *St6galnac1* KO | + | - | Our study' Fig. 5D–F | Single copy of *St6galnac1* | + | - | + |
| | *St6galnac1* WT | - | + | | Heterozygote triplication of *St6galnac1* | - | + | - |

rise to indirect genetic effects and greatly increases the total genetic variance of microbiome phenotypes. Failing to account for these indirect genetic effects produces underestimates of host genetic effects.

This is the largest ever characterisation of the rat caecal microbiome. In the human colon, *Lachnospiraceae* is the most abundant bacterial family, followed by *Bacteroidaceae*, *Oscillospiraceae* (formerly *Ruminococcaceae*) and *Bifidobacteriaceae*[48]. In the rat caecum, *Lachnospiraceae* was also the most abundant family, followed by *Bacteroidaceae*, *Oscillospiraceae* and *Muribaculaceae* (formerly *S24-7*). Thus, human and rat microbiomes are similar in terms of the most abundant bacterial families.

We uncovered 214 significantly heritable microbiome phenotypes (39.2% of microbiome phenotypes considered, FDR < 0.1), providing evidence that a large fraction of prevalent microbiome phenotypes are affected by host genetic effects. The large sample size we used ($N = 3784$) and the limited environmental (experimental) variation in our study likely explain why we detected so many heritable microbiome phenotypes. More specifically, some relevant factors did not vary at all (e.g. diet) while others did vary but did not contribute to the phenotypic variance (e.g. time of day, because all samples were collected at roughly the same time of day). A recent study of wild baboons that leveraged high-quality records of environmental variables and longitudinal data to account for environmental variation uncovered significant host genetic effects on 97% of prevalent microbiome phenotypes analysed[49]. Thus, it is increasingly clear that, across species, host genetic effects play an important role and that the substrate exists for hosts to evolve in response to selection on microbiome phenotypes.

Host genetic effects for the 214 significantly heritable phenotypes explained an average of 3% (s.d. 1.2%) of phenotypic variation. Since heritability always depends on the study design, we used two approaches to put our results into perspective. First, we compared heritability to the magnitude of maternal and cage effects. Host genetics were just as important for microbiome phenotypes as maternal effects, while cage effects were larger. We also compared the heritability of microbiome phenotypes to the heritability of organismal phenotypes measured in the same rats, which showed that microbiome phenotypes have generally lower heritabilities.

Our study is the first to examine the degree of consistency of polygenic host genetic effects across environments (cross-environment genetic correlations). We found consistent host genetic effects across cohorts (i.e., high cross-cohort genetic correlations). This result is consistent with a previous cross-cohort analysis of the effect sizes of GWAS-significant loci in humans[11] and suggests that statistical power is the main limitation for replicating microbiome-associated loci in humans as in other species.

Remarkably, we detected three microbiome-associated loci replicated in three or four cohorts of HS rats. At one locus, significantly associated with *Paraprevotella* in all four cohorts of HS rats, we identified a likely causal variant: a 9 kb copy number variant segregating both in HS and wild rats and affecting *St6galnac1*, the main gene responsible for the sialylation of gut mucins, a process known to play a key role in host-microbiota interactions[36]. Notably, this copy number

variant was also associated with *A. muciniphila* and *M. intestinale* in HS rats (not genome-wide significant) and a previous study by Yao et al. had reported significant differences in the abundance of these two bacteria between *St6galnac1* knock-out (KO) and wild type (WT) mice[36], providing strong evidence that the *St6galnac1* copy number variant indeed underlies the associations observed in HS rats. Comparing the abundances in the study by Yao et al. and ours (Table 1) suggests that the triplication of *St6galnac1* in HS rats results in higher gene expression, as might have been expected.

How, then, might the sialylation of gut mucins affect the abundance of *A. muciniphila*, *M. intestinale* and *Paraprevotella*? The study by Yao et al. demonstrated that *St6galnac1* is involved in the terminal sialylation of O-linked and N-linked mucin glycans and that *A. muciniphila* needs to remove this terminal sialyl moiety before it can feed on the glycans, which explains the association between this bacterium and *St6galnac1*. Another study has shown that the sialic acid released by *A. muciniphila* supports the growth of other, cross-feeding bacteria[50], which suggests *M. intestinale* might cross-feed with *A. muciniphila* in a way that depends on *St6galnac1* activity. Finally, the similar effect, in HS rats, of the *St6galnac1* variant on *A. muciniphila* and *Paraprevotella,* suggests that *Paraprevotella* may also be a mucin degrader capable of desialylating mucin glycans before feeding on them. A previous study aimed at identifying mucin degraders failed to identify *Paraprevotella* as such, but it also failed to identify *A. muciniphila*, despite the well-established mucin-degrading capacities of this bacterium[51]. Thus, additional work is required to determine if *Paraprevotella* is also a mucin degrader and in which conditions (HS rats were fasted in three of the four cohorts we studied).

In humans, we found evidence suggesting that *Paraprevotella* abundance is associated with *ST6GAL1*, a gene involved in the sialylation of mucin N-glycans. These findings point to shared modalities of host genetic effects on the microbiome in rats and humans, and they add to the evidence showing that host glycans play an important role in host-microbiome interactions[52–54].

Our findings are important for the following additional reasons. First, they suggest a causal mechanism behind the increase in *Paraprevotella* in patients with immunoglobulin A nephropathy (IgAN)[55,56] and the association between *ST6GAL1* and IgAN[57]. IgAN is an autoimmune disease in which deglycosylated IgA molecules cross the intestinal epithelium into circulation, where they are recognised by IgG autoantibodies, forming immune complexes that deposit in the kidneys. *A. muciniphila* is capable of desialylating and, more generally, deglycosylating IgA, which induces IgAN[58]. Our results showed that variants in *ST6GAL1* affected the abundance of *Paraprevotella* in the human gut, and they suggested that *Paraprevotella* has similar desialylating and deglycosylating activities as *A. muciniphila* in the rat gut. We therefore hypothesize that *Paraprevotella* can desialylate and deglycosylate IgA and induce IgAN. This effect would be magnified by the coating of *Paraprevotella* with IgA, which results from *Paraprevotella*'s ability to induce trypsin autodegradation and protect gut IgA from degradation by trypsin[59].

Secondly, our results suggest that *Paraprevotella* might mediate the association between *ST6GAL1* and breakthrough SARS-CoV-2 infection[60], through two distinct mechanisms. First, *Paraprevotella*'s

trypsin-degrading activity reduces the entry of coronaviruses into the host's gut epithelial cells[59] and, secondly, *Paraprevotella*'s IgA-protective effect potentiates vaccine efficiency, as demonstrated, in mice, for a gut pathogen[59]. If *Paraprevotella*, an anaerobic gut bacterium, indeed influenced breakthrough SARS-CoV-2 infection, it would suggest that these breakthrough infections originate from the gut, in other words, a role for faecal-oral transmission of SARS-CoV-2[61].

Given the evidence reported above and elsewhere that loci underlying host genetic effects, in particular genes responsible for glycan biosynthesis, play a role in susceptibility to pathogens[52,54,62] and given recent evidence of microbiome transmission in humans[16–18], it is clear that modelling approaches that fully account for microbiome transmission in the population are necessary to understand the evolution of host-microbiome interactions and how they relate to disease. This is most obvious for infectious diseases, an area where IGE models are under active development[46,63], but also for the broad range of diseases influenced by the microbiome[16,64,65]. Our detection of significant Mic-IGE demonstrates that some bacteria are both affected by host genetic effects and horizontally transmitted. Using well-established theory from the field of IGE[45], we showed that Mic-IGE can contribute substantially to the total genetic variance of microbiome phenotypes and that classical estimates of host genetic effects can greatly underestimate the total genetic variance of microbiome phenotypes. To the extent that microbiome traits influence disease, these conclusions will extend to disease phenotypes.

Finally, using simulations, we showed that ignoring IGE can result in underestimating DGE (i.e., heritability). DGE will be underestimated when (distantly) related individuals are included in the sample, IGE occur only between unrelated individuals, and DGE and IGE are positively correlated. Because the UK Biobank and other population-based cohorts include distantly related individuals[66], this issue could affect a large number of estimates from these cohorts and contribute to the issue of "missing heritability"[67,68]. For example, phenotypes measured in the UK Biobank that are affected by spousal IGE and for which DGE and spousal IGE are positively correlated[47] are expected to be affected.

Our study has limitations. First, 16S data do not allow for precise characterisation of the functional profile of the microbiome. A comparison of hundreds of mouse and human microbiomes has revealed that microbial functions rather than microbial species are conserved between the two host species[69]. Hence, to fully exploit HS rats as a model for humans, it will be important to characterise the microbiomes of HS rats at the functional level. Secondly, our results (QQ plot, Fig. 5B) suggest we had limited power to detect polygenic Mic-IGE, and we were clearly underpowered to map Mic-IGE. In the future, additional HS rat cohorts can be profiled to overcome this issue, as this population is actively used by the research community, and caecal samples are easy to collect and share.

## Methods
### Animals
All procedures were approved by the University at Buffalo, the University of Michigan and the University of Tennessee Institutional Animal Care and Use Committees. The rat Heterogeneous Stock (HS) colony was established at the NIH in 1984 from eight inbred strains: ACI/N, BN/SsN, BUF/N, F344/N, M520/N, MR/N, WKY/N, and WN/N population founded at the NIH[23] and thereafter maintained using an outbreeding scheme. The rats included in this study all originate from the HS colony maintained at the Medical College of Wisconsin (official nomenclature: MCW: NMcwi:HS, Rat Genome Database ID: 2314009) and are from generations 73 to 80. At weaning, the rats were shipped to one of three facilities, in the states of Michigan (MI), New York (NY) and Tennessee (TN). The rats that arrived in MI and NY (MI and NY cohorts) had their behaviour profiled. The rats that arrived in TN (TN2 cohort) were bred for a single generation to produce offspring (TN1), whose behaviour was profiled. Males and females were included in

about equal proportions in all cohorts. All protocols were approved by the Institutional Animal Care and Use Committees for each of the relevant institutions.

### Experimental design and how it relates to confounding between host genetic effects (=Mic-DGE), Mic-IGE, cage and maternal effects
**Breeding**. Typically, one male and one female from each litter born at the Medical College of Wisconsin were used to breed the next generation of rats at MCW. Another male and another female from the litter were selected for shipment to MI, and the same for shipments to NY and TN. The rats shipped from MCW to TN formed the TN2 cohort, but they were also used to breed the rats in the TN1 cohort (Fig. 1). The breeding pairs at MCW and in TN typically had only one litter.

**Co-housing**. Siblings were never co-housed in the MI, NY and TN2 cohort, but same-sex siblings were co-housed in the TN1 cohort.

**Confounding from cage and maternal effects**. Related individuals often share an environment. To the extent that this environment affects the phenotype studied, there will be confounding between genetic and family environmental effects[70]. If the relevant environmental factors have been recorded, they can be accounted for in the models, and unbiased genetic estimates can be obtained.

In our study, full-siblings and more distantly related individuals were included, both within and between cohorts. The relevant family environmental factors to consider, for microbiome phenotypes and in our laboratory setting, are maternal factors (e.g., maternal microbes that can be transmitted to the offspring in early life, maternal milk composition) and, when siblings are co-housed (TN1 cohort only), micro-environmental factors that affect all the rats in a cage in the same way but affect rats in different cages differently (e.g., chance exposure to facility microbes, noise levels).

**Confounding from cage effects**. Since siblings were not co-housed in the MI, NY and TN2 cohorts, we do not expect confounding of host genetic effects (Mic-DGE) by cage effects in these cohorts (i.e., confounding avoided *by design*). However, even when related individuals are not co-housed, Mic-IGE can still be partly confounded with cage effects: when the number of rats in the cage is greater than two. Indeed, in that case, any two focal rats will have at least one cage mate in common, and IGE from this or these shared cage mate(s) will affect the two focal rats in the same way, resulting in partial confounding with cage effects. As there were 3 rats per cage in the MI cohort and 2–4 rats per cage in the TN1 cohort, there was partial confounding between Mic-IGE and cage effects in these cohorts. In addition, in the TN1 cohort, same-sex siblings were co-housed, so there was confounding between host genetic effects/Mic-DGE, Mic-IGE and cage effects. To control for confounding for cage effects, we fitted a random term for cage effects (based on cage identity) in all our genetic analyses. We used the same model with cage effects for all cohorts.

**Confounding from maternal effects**. Genetic and maternal effects are necessarily confounded to some extent. In our study, the use of multiple offspring from each mother permits distinguishing between genetic effects and maternal effects, and we accounted for maternal effects in all our genetic analyses.

The models are described in more detail below.

**Evidence that our genetic estimates are unbiased and our p-values calibrated**. We previously showed in an extensive simulation study based on a dataset where relatives were co-housed (the greatest risk of confounding) that fitting cage effects jointly with polygenic DGE and IGE yielded unbiased estimates of polygenic DGE and IGE (Supplementary Fig. 6 and Supplementary Fig. 7 in Baud et al.[43]).

**Table 2 | Sample randomisation**

| Qiita prep ID: | 4075 | 4078 | 6462 | 6826 | 6827 | 6935 | 7783 | 7808 | 7831 |
|---|---|---|---|---|---|---|---|---|---|
| Qiita prep year: | 2017 | 2017 | 2019 | 2019 | 2019 | 2019 | 2019 | 2019 | 2019 |
| Sample collected: 2015 | 141 | 101 | 0 | 7 | 0 | 0 | 27 | 29 | 25 |
| Sample collected: 2016 | 458 | 253 | 6 | 64 | 34 | 34 | 33 | 44 | 23 |
| Sample collected: 2017 | 0 | 160 | 99 | 206 | 267 | 96 | 58 | 56 | 55 |
| Sample collected: 2018 | 0 | 0 | 45 | 140 | 168 | 118 | 52 | 39 | 81 |
| Unknown collection date | 13 | 5 | 18 | 77 | 59 | 22 | 362 | 362 | 340 |

In the current study:
- Supplementary Fig. 12 shows that Mic-IGE $p$-values follow the expected uniform distribution when Mic-DGE, cage and maternal effects, but no Mic-IGE are simulated.
- Figure 6C and Supplementary Fig. 13 show that permuting cage mates' assignments yields significantly lower estimates of total genetic variance, Mic-IGE and Mic-DGE.
- Figure 6D and Supplementary Fig. 14 (green boxplots) show, based on simulations, that DGE (e.g., host genetic effects) are unbiased by cage in the NY and MI cohorts.

## Housing conditions
The type of cages used and pathogen status of the various facilities varied: at the Medical College of Wisconsin, the rats were housed in ventilated cages in an SPF facility; in MI, the rats were housed in conventional cages in a non-SPF facility; in NY, they were housed in conventional cages in an SPF facility; finally, in TN, the rats were housed in conventional cages for those rats studied before March 2016 and in ventilated cages after March 2016, always in SPF facilities. The number of rats co-housed in a cage also varied between the cohorts: in the NY and TN2 cohorts, the rats were housed in groups of 2; in MI, in groups of 3, and in the TN1 cohort, in groups of 2 or 3. All rats were on a chow diet but the food vendors varied across the different facilities: at the Medical College of Wisconsin, the rats were fed with Teklad 5010 diet ad libitum (Envigo, Madison, Wisconsin); at the University of Michigan, they were fed irradiated Picolab Laboratory Rodent Diet (LabDiet, St. Louis, Missouri); at the University at Buffalo, Teklad 18% Protein Rodent Diet (Envigo); at the University of Tennessee Health Science Center, Teklad Irradiated LM-485 Mouse/Rat Diet (Envigo).

## Phenotyping
The behaviour of the rats in the NY, MI, and TN1 cohorts was profiled using different assays that focused on traits relevant to substance (cocaine and nicotine) use disorder in humans[71,72]. In addition, physiological measures were collected in all four cohorts, mostly from tissues collected after euthanasia. The rats from the MI, NY and TN2 cohorts, but not the rats from the TN1 cohort, were fasted overnight before euthanasia and tissue collection. For euthanasia, in NY and TN, the rats were anaesthetised with isoflurane before being quickly decapitated. In NY and MI, the rats were euthanised at 6.5 and 3 months of age, respectively. TN1 rats were on average 2 months old when their ceca were collected, and TN2 rats were 6 months old.

## Isolation of the caecal microbiome
The ceca were dissected from each rat within minutes of euthanasia. The caecum was removed by cutting immediately at the entrance of the small intestine (ileum) to the caecum and immediately below the exit of the caecum to the large intestine (colon). The caecum was placed into a sterile container and stored at −80 °C. The caecal content for microbiome analysis was taken from the apical part of the ceca. Ceca were thawed on ice to the extent that they could be cut, cut in half using a clean razor blade, and ~100 μL of content (without host tissue) was scooped and transferred into a 1.5 ml tube and frozen on dry ice.

## Sample randomisation, DNA extraction, library preparation and sequencing of the 16S amplicon
Sample randomisation was carried out by mixing a few hundred of the samples available at the time of processing (see the table below) in a very large polystyrene box, returning the samples to sample boxes and processing the sample boxes in a random order.

As shown in Table 2, two batches of samples were processed in 2017 (one batch including randomised samples collected in 2015 and 2016, the other batch including randomised samples collected in 2015, 2016 and 2017), and the remaining seven batches were sequenced in 2019 (including randomised samples collected 2015–2018).

DNA was extracted from the caecal content within a few hours of the caecal dissection, using the MoBio (now Qiagen) PowerSoil DNA high-throughput isolation kit as per the manufacturer's recommendations.

The V4 region of the 16S SSU rRNA gene was amplified using the updated, barcoded primers used in the Earth Microbiome Project (https://earthmicrobiome.org/protocols-and-standards/16s/). 5–10% PhiX was added to each library. The libraries were run in eleven batches on a MiSeq (Illumina) machine to obtain 150 bp paired-end reads (average library depth 20,842 pairs, minimum: 6624, maximum: 34,495).

## Sample filtering
Negative controls and rats with unreliable metadata or no genotypes were excluded from subsequent analyses. In addition, we excluded samples with too few or too many sequencing reads (mean ± 2× standard deviation), as extreme variation in library size could be problematic for compositional data analysis, even when statistical methods are used to account for it (see the section on CLR transformation below). This left a total of 3897 rats across the four cohorts.

## 16S data processing and identification of amplicon sequence variants (ASVs)
Raw sequencing data were first demultiplexed, and quality control was carried out using split_libraries_fastq.py (qiime2 version 1.9.1). This resulted in a median number of reads per sample of 20.9 K. Secondly, reads were trimmed to length 150 bp and adaptor sequences removed using Trimming.py (qiime2 version 1.9.1). ASVs were identified using Deblur (2021.09 workflow) and Greengenes 1 ("Greengenes 13.8") as reference. At this point, the deblur reference hit tables were chosen. ASVs were filtered using filter_features.py (qiime2 version 2023.2.0) against the Greengenes 2 ("Greengenes 2022.10") reference. Subsequently, these data objects were merged to yield microbiome profiles with 93,090 ASVs for 4422 samples, including 268 negative controls (30 samples with too few reads were dropped by the Deblur workflow).

## ASV taxonomy assignment
We assigned taxonomy to each ASV using the latest Greengenes 2 taxonomy (downloaded from http://greengenes.microbio.me/greengenes_release/2022.10-rc1/2022.10.taxonomy.asv.nwk.qza), using the following qiime command line:

qiime greengenes2 taxonomy-from-table --i-reference-taxonomy 2022.10.taxonomy.asv.nwk.qza --i-table 175568_feature-table.qza --o-classification biom.taxonomy.qza.

## Average microbiome profiles

Average microbiome profiles were calculated for each cohort separately, after collapsing ASVs at the family level (summing up their relative abundances).

## Rarefaction of the counts table (for alpha diversity)

For calculating alpha diversity (Supplementary Fig. 2) only, the count matrix (analysis ID 57950 in Qiita, artefact ID 175568) was rarefed to 10,000 reads in Qiita in 2728 samples (artefact ID 214605).

## Alpha diversity

Within-sample ("alpha") diversity was calculated after rarefying the count table and collapsing ASVs at the family and genus levels. Richness was calculated as the number of observed families or genera in a sample.

## Centred log-ratio (CLR) transformation of raw ASV abundances (for beta-diversity and all genetic analyses)

To account for the compositional nature of the data[29], we transformed the raw abundance values of each ASV using the CLR transformation (null values were replaced by 0.00001). CLR-transformed values are to be interpreted as relative abundances with respect to the geometric mean of the sample.

## Beta-diversity

Between-sample ("beta") diversity across all rats was calculated from the CLR-transformed ASV abundance values collapsed at the family level, as the Euclidean distance between samples. This distance is equivalent to the Aitchison distance[29]. Figure 2B shows the first two principal components of the CLR-transformed, family-level abundance table.

## Filtering of microbiome phenotypes based on prevalence

For all genetic analyses, to ensure sufficient precision of the genetic estimates and statistical power to detect associated loci, and to limit issues arising from zeroes, we only considered ASVs and taxa present in at least 50% of the rats in a given cohort (MI, NY, TN1, TN2 or whole sample).

## Across-samples normalisation of microbiome phenotypes and regression of the effects of experimental covariates

For all genetic analyses and to satisfy the assumption of normality made by the linear mixed models we used for all the genetic analyses, each microbiome phenotype was quantile (i.e., rank) normalised, with ties broken at random. We also regressed out any potential effect of sex, phenotyping batch, 16S library preparation batch, library size, and, for the TN1 and TN2 samples, weaning suite, using fixed effects in linear models. We used the residuals of these models for genetic analysis.

## Genotyping

Genomic DNA was isolated from spleen samples using the Beckman Coulter DNAdvance Kit (Beckman Coulter Life Sciences, Indianapolis, IN, USA). Genomic library preparation, sequencing, and genotyping were all conducted using the pipeline described by Chen et al.[73]. Briefly, genomic samples were prepared using both double-digest genotyping by sequencing (ddGBS) and low-coverage whole genome sequencing (lcWGS) techniques. ddGBS libraries were produced following Gileta et al.[74] and sequenced on an Illumina HiSeq 4000 DNA sequencer (Illumina, San Diego, CA, USA) using 100-bp single-end reads. lcWGS libraries were produced following Chen et al.[73] and

sequenced on an Illumina NovaSeq 4000 using 150-bp paired-end reads. All libraries were sequenced at the UC San Diego Institute for Genomic Medicine Genomics Center.

Sequenced libraries were demultiplexed using fastx_toolkit v0.0.14 (for ddGBS samples)[75] and fgbio v1.3.0 (lcWGS)[76], trimmed using cutadapt v4.4 (ddGBS, lcWGS)[77] and bbDuk v38.94 (lcWGS)[78], and aligned to the rat reference genome mRatBN7.2 (rn7) from the Rat Genome Sequencing Consortium (NCBI RefSeq assembly GCF_015227675.2)[79] using bwa-mem v0.7.17[80]. Mapped sequences were used to impute SNP genotypes using STITCH v1.6.6[81]. From this set, we removed all SNP loci with low imputation quality scores produced by STITCH v1.6.6 (INFO < 0.9). Genotypes for the 7,358,643 SNPs passing this filter are available at https://library.ucsd.edu/dc/object/bb29129987. This dataset includes many more rats than the rats used in this study. The files downloaded from this URL were renamed as "P50_Rn7" (see below).

For all analyses in this study, we focused on SNPs that were on the autosomes (chromosomes 1–20 in the rat). For constructing genetic relatedness matrices and for GWAS, but not for the LocusZoom plots, we additionally removed loci with high missing data (missing rate >0.1) or low minor allele frequencies (MAF < 0.05), leaving 4,880,609 autosomal SNPs. For GWAS, missing genotypes at these 4,880,609 SNPs were mean imputed.

## Construction of the SNP-based genetic relatedness matrices (GRMs)

For the heritability and genetic correlation analyses, a GRM was constructed from a subset of pruned SNPs on autosomes 1–20. For GWAS, 20 different GRMs were constructed for use in the leave-one-chromosome-out (LOCO) framework[82–84], using the same subset of pruned SNPs but excluding SNPs on one chromosome at a time.

We used PLINK v1.90b6.2[85] to prune SNPs and create the GRMs:
#prune:
plink --bfile ./P50_Rn7 --indep 50 5 2 --out ./P50_Rn7_pruned_50_5_2
We used the following command to create the GRM from all SNPs on chromosomes 1–20
#create GRM from SNPs on chromosomes 1–20:
plink --bfile ./P50_Rn7 --make-rel square --keep ./P50_Rn7_rats2include.txt --chr 1-20 --extract ./P50_Rn7_pruned_50_5_2.prune.in --maf 0.05 --geno 0.1 --out ./P50_Rn7_pruned
#create LOCO GRMs for chr in {1..20}:
plink --bfile ./P50_Rn7 --make-rel square --not-chr $chr 23 24 26 --keep ./P50_Rn7_rats2include.txt --extract ./P50_Rn7_pruned_50_5_2.prune.in --maf 0.05 --geno 0.1 --out ./P50_Rn7_pruned_LOCO_${chr};

## Analyses of polygenic host genetic effects

We estimated the magnitude of polygenic host genetic effects (SNP heritability) using the following linear mixed model:

$$y = Xb + a + e + Wc + W'm \qquad (1)$$

where y is the vector of phenotypic residuals, $X$ is a vector describing how many rats are in each cage and $b$ the corresponding estimated fixed effect, a is a vector of random additive genetic effects, $e$ is a vector of random non-genetic effects, W is the matrix of cage assignments and $c$ the corresponding vector of random cage effects. $W'$ is the matrix relating the phenotyped individuals to their mothers and m the corresponding vector of random maternal effects. a follows a multivariate normal distribution $N(0, A)$, where $A$ is the GRM (see the section above). $e$ follows a multivariate normal distribution $N(0, I)$, where $I$ is the identity matrix.

Due to missing information for a few cages, the final sample sizes in all genetic analyses were 1107 for the MI cohort, 1166 for the NY cohort, 941 for the TN1 cohort, and 553 for the TN2 cohort (3767 for the joint sample).

All models were fitted via restricted maximum likelihood using the LIMIX software[30,86] and a new covariance class was implemented to fit bivariate models. The significance of heritability was assessed using a likelihood ratio test (LRT) with 1 degree of freedom, comparing model (1) to a reduced model without additive genetic effects a. To estimate the genetic correlation between the same microbiome phenotype measured in two cohorts, a bivariate version of model (1) was used. The significance of the genetic correlation was assessed using an LRT with 1 degree of freedom.

## GWAS

The null model for GWAS was model (1) modified to use the LOCO GRM matrices (see the section above). The alternative model included an additional fixed effect for the variant being tested. The null and alternative models were compared using an LRT with one degree of freedom (dof). The genome-wide significance threshold ($-\log P > 5.8$), which accounts for the number of independent SNPs tested, was calculated by permuting the rows (rats) of the genotype matrix 1000 times, running a GWAS on these permutations, recording the smallest $p$-value for each, and eventually calculating the 95th percentile of the distribution of these minima[30,31]. The adjusted significance threshold, which further accounts for the number of independent microbiome phenotypes considered in a cohort, was calculated by Bonferroni correction of the genome-wide significance threshold. This number was estimated from a principal component analysis of the microbiome phenotypes measured in the cohort, counting the number of eigenvalues necessary to explain 99% of the total variance. All models were fitted using the LIMIX software.

## Analysis of indirect genetic effects on microbiome phenotypes (Mic-IGE)

The following model, which is similar to the model used in two of our previous studies on IGE[42,43], was used to quantify indirect genetic effects:

$$y = Xb + a_D + Za_S + e_D + Ze_S + Wc + W'm \qquad (2)$$

$y$, $X$, $b$, $W$, $c$, $W'$ and $m$ are defined as in the previous section. $a_D$ is the vector of random additive direct genetic effects (Mic-DGE). $a_S$ is the vector of random additive indirect genetic effects (Mic-IGE) and $Z$ is the matrix indicating, for each rat, which are the cage mates (importantly $Z_{i,i} = 0$). $e_D$ and $e_S$, also random effects, refer to the non-genetic component of direct and indirect effects.

The joint distribution of all random effects was defined as:

$$\begin{bmatrix} \sigma_{A_D}^2 A & \sigma_{A_{DS}} A & 0 & 0 & 0 & 0 \\ \sigma_{A_{DS}} A^T & \sigma_{A_S}^2 A & 0 & 0 & 0 & 0 \\ 0 & 0 & \sigma_{E_D}^2 I & \sigma_{E_{DS}} I & 0 & 0 \\ 0 & 0 & \sigma_{E_{DS}} I^T & \sigma_{E_S}^2 I & 0 & 0 \\ 0 & 0 & 0 & 0 & \sigma_C^2 I & 0 \\ 0 & 0 & 0 & 0 & 0 & \sigma_M^2 I \end{bmatrix}$$

All models were fitted using the LIMIX software. The significance of Mic-IGE was assessed using an LLR test, comparing model (2) to a reduced model without Mic-IGE (i.e., $Za_S$) but with indirect environmental effects ($Ze_S$). Because Mic-DGE and Mic-IGE are correlated ($\sigma_{A_{DS}}$), the LLR statistics follow a mixture of $\chi^2$ distributions with degrees of freedom 1 and 2[87]. To determine which mixture parameter was appropriate, we used a parametric bootstrap approach, simulating phenotypes under the null hypothesis of no Mic-IGE, calculating $p$-values for them using an LLR test with mixture parameter varying between 1 and 2, and choosing the mixture parameter (0.7) that yielded calibrated $p$-values (see Supplementary Fig. 13).

The total heritable variance[88] was calculated as:

$$\sigma_H^2 = \sigma_{A_D}^2 + 2(n-1)\sigma_{A_{DS}} + (n-1)^2 \sigma_{A_S}^2 \qquad (3)$$

where $n$ is the average number of rats in a cage.

## Evaluation of the calibration of Mic-IGE p-values using null simulations (Supplementary Fig. 12)

Following a parametric bootstrap approach, we simulated phenotypes from the null model (model with DGE, cage and maternal effects but no IGE) fitted to the microbiome phenotype most significantly affected by Mic-IGE (ASV_13916_all). We analysed these null simulations using both the null model and the full model (with IGE), and to obtain one Mic-IGE $p$-value for each simulation, we carried out an LRT with a mixture of $\chi^2$ distributions with degrees of freedom 1 and 2, varying the mixture parameter between 0 and 1 (top left panel to bottom right panel)[89]. A mixture parameter of 0 (i.e., LRT with 1 degree of freedom) ignores the DGE-IGE covariance parameter ($\sigma_{A_{DS}}$). A mixture parameter of 1 (i.e., LRT with 2 degrees of freedom) assumes that the IGE parameter ($\sigma_{A_S}^2$) and the DGE-IGE covariance parameter ($\sigma_{A_{DS}}$) are independent (but they are not). We chose the mixture parameter that yielded the most calibrated $p$-values for these null simulations (0.7, highlighted with a red rectangle in Supplementary Fig. 12). With this mixture parameter, the $p$-values of null simulations are, indeed, calibrated (i.e., they follow the uniform distribution expected in the absence of IGE). We used this mixture parameter in all analyses of real and permuted data.

## Permutations of cage mates' assignments and evaluation of genetic variance components (Fig. 6C and Supplementary Fig. 13)

The cage mates' assignments, which are derived from the cage records and used to model IGE, were permuted, but the cage assignments, used to model cage effects, were not.

## Full simulations (with IGE) and evaluation of DGE when IGE are ignored (Fig. 6D and Supplementary Fig. 14)

To illuminate, using simulations, the impact on DGE estimation of not accounting for IGE where they are present, we simulated phenotypes from DGE, IGE and cage effects (DGE arbitrarily set to 0.4, IGE to 0.2, their correlation to 0.9; this corresponds to strong IGE; note that no maternal effects were simulated here for simplicity) and analysed these simulations with either a model including DGE, IGE and cage effects (green boxplots) or a model including only DGE and cage effects (blue boxplots).

## Alignment of wild rat sequences

Raw sequences from wild rats caught in various locations in China were downloaded from SRA with accession number SRP078989 (BioProject PRJNA330658)[32]. Trimming of adaptors was performed using TrimGalore[90]. The remaining reads were aligned to the rn7 reference genome using bwa-mem 0.7.17 with default parameters[80]. BAM files were sorted and filtered, retaining only reads with a mapping quality (MQ) $\geq 30$ using samtools 1.12[91]. Mate coordinates and sizes were fixed using samtools fixmate -m. Subsequently, duplicates were removed using samtools markdup -r -s. Variant calling was performed using GTAK 4.1.8[92]. Briefly, we called variants using HaplotypeCaller and used Base Quality Score Recalibration (BQSR) to recalibrate the base quality scores of BAM files. Germline calling was performed. Instead of using the VQSR step with polymorphisms from laboratory strains, we applied hard filtering of the final VCFs to avoid bias towards the variants present in the laboratory animals.

We lifted over the available public New York rat data, which used the rn5 rat genome assembly, to rn7 using UCSC chain files[93] and Picard tools 2.25.5 LiftoverVcf[94].

VCF files from wild Chinese rats, wild rats from NYC, and inbred rat strains were then merged using bcftools isec[91].

## Other bioinformatics tools

The Variant Visualizer tool from the Rat Genome Database[95,96] was used to identify variants segregating in the eight HS founders and SHRSP, and their functional consequences were predicted by Polyphen[97].

Plink[85] was used to calculate LD ($R^2$).

The standalone version of LocusZoom (version 1.3) was used.

The R code used to generate Supplementary Fig. 4 was adapted from the code published by Grieneisen et al.[49].

IGV[98] was used to generate Supplementary Fig. 8.

## Boxplots used for the visualisation of the data

In all boxplots, the thick bar shows the median of the data, the upper/lower hinges show the 1st and 3rd quantiles, the lower/upper whiskers show the values within Q1 - 1.5*IQR and Q3 + 1.5*IQR (; Q1 and Q3 are the 1st and 3rd quantiles, IQR = interquartile range), and all data points outside of this range (outliers) are plotted as dots. Only in Fig. 5D–F were outliers not plotted, for better visualisation of the quantiles (21, 20 outliers, and 4 outliers in genotype groups "Single copy", "Het", "Triplicated" in Fig. 5D; 42, 25, and 1 outliers for Fig. 5E; 34, 12, and 1 outliers in Fig. 5F).

## Reporting summary

Further information on research design is available in the Nature Portfolio Reporting Summary linked to this article.

## Data availability

All 16S sequencing data and metadata are available from Qiita[99] study ID 11479, ENA accessions ERP182733/ PRJEB101310 [https://www.ebi.ac.uk/ena/browser/view/ERP182733] and Figshare dataset "Host/microbiome interactions in NIH-Heterogeneous Stock rats (study based on 16S data)" [https://doi.org/10.6084/m9.figshare.28769039]. The genotype data are available from UCSD's Library Digital Collections under the title "Heterogeneous Stock (HS) Rat Genotypes, Version 4" [https://doi.org/10.6075/J0X63N54] (date of version: 2023-12-15; corresponding fastq files available from SRA BioProject PRJNA1022514). Wild rat from China raw sequences were downloaded from ENA BioProject PRJNA330658 (secondary study accession SRP078989)[32]. NYC metro rats genotypes were downloaded from the Dryad dataset "Genetic Adaptation in New York City Rats" [https://datadryad.org/stash/dataset/doi:10.5061/dryad.08kprr4zn][33]. Rat inbred strains vcf files were obtained from the Rat Genome Database "strain_specific_variants" dataset [https://download.rgd.mcw.edu/strain_specific_variants/Dwinell_MCW_HybridRatDiversityProgram/Jan2023/mRatBN7/]. The Illumina short reads used for Fig. 5b are available from SRA BioProject PRJNA1048943 (study SRP476247). Intermediate data files produced by and used in our analyses are available from the Figshare dataset "Host/microbiome interactions in NIH-Heterogeneous Stock rats (study based on 16S data)" [https://doi.org/10.6084/m9.figshare.28769039]. Source Data files for the data presented in graphs within the figures are available from the same Figshare repository [https://doi.org/10.6084/m9.figshare.28769039], in the form of one R object (.RData) per figure.

## Code availability

The R scripts used to generate all the results and Figures reported in the paper are available from https://github.com/Baud-lab/P50/tree/Master/16S (10.5281/zenodo.17457488)[100]. The README details the inputs and outputs, some of which are available from the Figshare repository as intermediate files. The extended LIMIX code used for the genetic analyses is available from https://github.com/Baud-lab/CoreQuantGen (10.5281/zenodo.4965824)[101].

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

## Acknowledgements

This work was supported by a Seed grant from the Centre for Microbiome Innovation, a Sir Henry Wellcome fellowship from the Wellcome Trust (105941/Z/14/Z to A.B.), a fellowship from La Caixa Foundation (ID 100010434) and from the European Union's Horizon 2020 research and innovation programme under the Marie Sklodowska Curie grant agreement No 847648 (LCF/BQ/PI21/11830006 to A.B.), a PRE2021-097413 contract funded by MCIN/AEI/10.13039/501100011033 and FSE+ to H.T., and two NIH/NIDA grants: P50DA037844 and U01DA051234, both to A.A.P. We acknowledge support of the Spanish Ministry of Science and Innovation through the Centro de Excelencia Severo Ochoa (CEX2020-001049-S, MCIN/AEI /10.13039/501100011033), the Generalitat de Catalunya through the CERCA programme and to the EMBL partnership. This publication includes genotype data generated at the UC San Diego IGM Genomics Center utilising an Illumina NovaSeq 6000 and NovaSeq X Plus that were purchased with funding from a National Institutes of Health SIG grant (S10OD026929). We thank the following members of the Knight lab: Greg Humphrey, MacKenzie Bryant, Karenina Saunders, Caitriona Brennan, and Rodolfo Saldo Bentiez for help processing the gut microbiome samples, and Gail Ackermann for help with metadata and processing. We thank the Rat Genome Database (RGD) team, particularly Jeffrey De Pons, for adapting some features of the Variant Visualizer to help our analyses.

## Author contributions

O.P., R.K., A.A.P. and A.B. designed the study. A.M.G., W.H., K.H., A.H., K.I., C.P.K., A.C.L., C.D.M., A.G.M., A.H.N., J.A.T., T.W., H.C., S.B.F., P.J.M., J.B.R., T.E.R., L.C.S.W., O.P., R.K., A.A.P. and A.B. contributed to, or supervised, the collection or processing of samples to obtain the microbiome data. A.G. and T.K. provided help with Qiita. H.T., D.C., F.M., J.G.-C., A.S.C., B.B.J., T.M.S., R.C., M.J.B., E.B., P.A.D., O.S., A.A.P. and A.B. contributed to, or supervised, the analysis of the genotype and microbiome data. A.A.P. and A.B. wrote the manuscript, with contributions and comments from many co-authors.

## Competing interests

R.K. is a scientific advisory board member and consultant for BiomeSense, Inc., has equity and receives income. He is a scientific advisory board member and has equity in GenCirq. He is a consultant for DayTwo and receives income. He has equity in and acts as a consultant for Cybele. He is a cofounder of Biota, Inc., and has equity. He is a cofounder of Micronoma and has equity and is a scientific advisory board member. The terms of these arrangements have been reviewed and approved by the University of California, San Diego, in accordance with its conflict-of-interest policies. The remaining authors declare no competing interests.

## Additional information

[1]Centre for Genomic Regulation, Barcelona Institute of Science and Technology, Barcelona, Spain. [2]Universitat Pompeu Fabra, Barcelona, Spain. [3]Department of Psychiatry, University of California San Diego, La Jolla, CA, USA. [4]Institute of Evolutionary Biology (CSIC-UPF), Department of Medicine and Life Sciences, Universitat Pompeu Fabra, Barcelona, Spain. [5]Department of Genetics, University of Groningen, University Medical Center Groningen, Groningen, the Netherlands. [6]Department of Pediatrics, University of California San Diego, La Jolla, CA, USA. [7]Clinical and Research Institute on Addictions, University at Buffalo, Buffalo, NY, USA. [8]Department of Pharmacology, Addiction Science and Toxicology, University of Tennessee Health Sciences Center, Memphis, TN, USA. [9]Department of Physiology, Medical College of Wisconsin, Milwaukee, WI, USA. [10]Department of Psychology, University of Michigan, Ann Arbor, MI, USA. [11]Department of Pharmacology and Toxicology, University at Buffalo, Buffalo, NY, USA. [12]Department of Psychology, University at Buffalo, Buffalo, NY, USA. [13]Center for Human Genetics, Institute of Molecular Medicine, McGovern Medical School, University of Texas at Houston, Houston, TX, USA. [14]Genome Biology Unit, European Molecular Biology Laboratory, Heidelberg, Germany. [15]Division of Computational Genomics and System Genetics, German Cancer Research Center, Heidelberg, Germany. [16]Michigan Neuroscience Institute, University of Michigan, Ann Arbor, MI, USA. [17]Department of Psychiatry, University of Michigan, Ann Arbor, MI, USA. [18]Department of Internal Medicine, Section on Molecular Medicine, Wake Forest University School of Medicine, Winston Salem, NC, USA. [19]Department of Computer Science & Engineering, University of California San Diego, La Jolla, CA, USA. [20]Shu Chien-Gene Lay Department of Bioengineering, University of California San Diego, La Jolla, CA, USA. [21]Halıcıoğlu Data Science Institute, University of California San Diego, La Jolla, CA, USA. [22]Center for Microbiome Innovation, La Jolla, CA San Diego, USA. [23]Institute for Genomic Medicine, University of California San Diego, La Jolla, CA, USA. [24]These authors contributed equally: Hélène Tonnelé, Denghui Chen. ✉e-mail: aap@ucsd.edu; amelie.baud@crg.eu

