## [Transparent Peer Review file · Nature Communications]

Genetic architecture and mechanisms of host/microbiome interactions from a multi-cohort analysis of outbred laboratory rats

Corresponding Author: Dr Amelie Baud

Version 0:

Reviewer comments:

Reviewer #1

(Remarks to the Author)

Review of Tonnele et al 2025

The manuscript by Tonnele and colleagues investigates the genetic architecture of microbiome composition in a very large cohort of heterogenous rats. The authors identify several genotype-microbiota associations that are investigated at the locus level with functional candidate genes names. Additionally there is considerable analysis of cage, maternal and "indirect" genetic effects. All of these are important aspects of the study. Below are some suggestions and questions in regards to the analysis and conclusions

Microbiome questions

- What was the distribution of zero inflated data in each cohort? How were 0 data accounted for? Use of high 0 inflation with rank normalization could be problematic in a linear model. This is most evident in supplementary figure 5 and figure 3. There is not a relationship between relative abundance and H2. There are heritable microbiome (ASV?) that are low abundance that also have low prevalence. The latter indicate relatively rare ASV are included in the results. Figure 3 indicate that most of the ASV are low Heritability and low prevalence and low abundance. For the ASV highlighted as genetically regulated, what is prevalence and relative abundance?
- What was the total number of specific sequences identified at a phylogenetic level (ie family) for the whole study? Did this vary between sites?
- This is a large study. How were samples randomized into sequencing runs?

Rat cohort questions

- The protocols at each site were different. In supplemental table 1 details of different protocols are listed. The authors should be more transparent about the variation in diet, drug exposure, housing conditions and phenotyping which all could affect results. Do the results indicate that the loci identified are invariant to environmental changes?
- The indirect genetic analysis is intriguing. The figure indicates a "genotype" effect but I interpret this to a "rat strain" effect as these are complex 8 way cross. The genotype effect would be more appropriate for a specific test such as a knockout animal. The model and results indicate that the effects of horizontal transmission (ie a within cage effect) could be dependent on genotype. It is not clear how robust these measures are and if these are random within cage effects.
- Additional question on maternal effects. Did the authors quantitate the maternal microbiome? It is difficult to understand the seeding or transmission of the microbiota without an actual measurement. The model described in the methods uses W' to model maternal effects. It isn't clear how many offspring per dam were included in the analysis. Are dams bred multiple times? Given the size of the study, it could be assumed "yes" but clarity is appreciated. If bred multiple times did the sires change between breeding's? How are half sibs and siblings from different generations dealt with in the analysis of direct and indirect genetic effects?
- Did littermates get sent to different locations?
- The polygenic host genetic effects also include a cage effect term (which should be confounded with maternal effects?)

Analysis

- Locus zoom (supplementary figure 10) indicate a genotyping gap on chr 10 at 102-103 Mb. Is there a technical reason for the gap?

- The simulations are difficult to understand. In supplemental figure 14 the authors state- “The reason why (A) and (B) are different is most likely that, in the NY cohort, the rats were housed 2 to a cage, hence had no shared cage-mate.” However, the figure panels look very similar. Can they clarify specific differences?
- The abstract focuses on ST6GAL1 and immunity. Do the authors have any data from these animals to support this statement—“this suggests ST6GAL1’s effects on IgA nephropathy and COVID-19 breakthrough infections may be mediated by Paraprevotella.” The genotype- microbiota association is strong but has not been tested in this paper. If data is not available, please remove from the abstract as this is not directly supported by the data in the current manuscript.

(Remarks on code availability)

Reviewer #2

(Remarks to the Author)

This manuscript by Tonnele, Chen, and colleagues analyzes host genetic influences on the gut microbiome in several cohorts of genetically diverse rats. This work addresses the open question of just how much host genetics influence the gut microbiome. One end of the spectrum is that microbiome heritability is nearly universal, but we need very large sample sizes and longitudinal data to detect this heritability (Grieneisen et al. 2021). The other end of the spectrum is that microbiome heritability is generally quite modest (Rothschild et al. 2018, Gacesa et al. 2022).

This paper provides support for appreciable microbiome heritability. Finding microbiome-associated loci that are consistent across cohorts is a compelling argument for a real genetic signal. Relating the hit involving Paraprevotella to human data was a nice connection.

Overall, the analysis was thoughtful, the methods were clear, and in general, the claims were supported by the data presented.

I admit that I remain skeptical about the final section relating to indirect genetic effects, namely that accounting for IGEs can increase the amount of explained genetic variance so dramatically (Fig. 6C). One idea for increasing confidence in this analysis is to repeat it using scrambled cage assignments. If IGEs fail to explain any additional variance when the model uses incorrect cage labels, I would find it reassuring that the methodology is not artificially inflating the amount of explained genetic variance. I believe this is different from the simulations shown in Fig. 6D. This analysis is not mission-critical.

One question that remains for me (and does not need to be addressed by the authors, though it would be interesting to hear their thoughts) is how to reconcile microbiome heritability with how dynamic we know the microbiome can be. In other words, we know that the microbiome changes day-to-day and week-to-week in response to factors like diet, medications, whom we live with, etc. And yet heritability implies stability, i.e. (fixed) host genetics has some stable effect on the microbiome (although one could imagine temporally dynamic genetic influences). This question could be explored with longitudinal data but unfortunately not in this dataset.

I have no major concerns and commend the authors for a solid manuscript. Please see below for minor concerns and additional comments.

Minor Concerns

- Line 60: “Given Paraprevotella’s known...” Interesting, but perhaps too speculative for the abstract (see comment about line 497 below).
- Lines 80-82: “Only two microbiome-associated...” It would be nice to include citations for this sentence.
- Line 183: “behavioural and physiological phenotypes”. Like what? It would be helpful to list a few examples of these phenotypes.
- Fig. 3B: why are there dots with prevalence below 0.5? I thought only ASVs with prevalence > 0.5 were tested (lines 173-174). Also, this plot is okay - but wouldn’t it be more straightforward to just plot median abundance versus heritability?
- Fig. 3D: I think this is a very important panel. My interpretation is that, in general, there is little concordance in heritability across cohorts (even though the authors point out that certain pairwise correlations are significant). Alternatively, one could argue that the global concordance may not be great, but there is concordance for the hits discussed in Fig. 4. Could the authors label these hits (from Fig. 4) in this panel?
- Line 260: “This analysis did not...” It might be nice to include a figure panel showing this point.
- Fig. 5D-F: I don’t follow. If the abundance of these ASVs is associated with St6galnac1, don’t we expect to see higher ASV abundance with duplications or triplications? At best, this is true only in panel F. Also, what is the difference between “single copy” and “het”?
- Lines 432-434: “We demonstrated that host genetic effects...” I agree that the 3 loci look real, but the heritability scatterplots (Fig. 3D) argue against the first half of the sentence. I know the authors also use genetic correlation to argue for concordant genetic effects, but it’s hard for me to understand high genetic correlations without concordant heritability estimates.
- Lines 484-486: “Together with evidence...” The argument is that Paraprevotella makes IgAN worse. Any thoughts about why IgAN patients have more Paraprevotella in the first place? Because of mutations in ST6GAL1?
- Line 497: “Thus, our results support...” Interesting but highly speculative. Seems like this claim is supported by evidence external to this study, namely that Paraprevotella degrades trypsin (leading to decreased viral entry into epithelial cells) and that Paraprevotella may improve vaccine efficiency. It feels like this summary sentence is just a little too far afield from the

findings actually presented in the paper, but maybe it's okay since it's in the discussion.

- Line 687: "each microbiome phenotype was quantile (i.e. rank) normalised, with ties were broken at random." This could be a problem when there are many zeros. Were they many zeros?

Praiseworthy

- Lines 172-173: "We analysed the different cohorts separately..." Analyzing the cohorts separately (before analyzing them jointly) was nice. Cross-cohort consistency is a compelling way to increase one's confidence in the hits.
- Lines 179-182: "Hence, the heritability of microbiome..." Definitely agree! This was a good analysis.
- Lines 184-185: "ASVs tended to have higher heritability..." Nice result.
- Line 339-342: "In a human cohort..." Nice connection to human data.

(Remarks on code availability)

I took a look through the Github for reproducing figures. It seemed reasonably thorough and well-organized. I did not try running any code.

Version 1:

Reviewer comments:

Reviewer #1

(Remarks to the Author)

I appreciate the careful responses and all previous queries have been addressed

(Remarks on code availability)

Reviewer #2

(Remarks to the Author)

The authors addressed my concerns and questions. In particular, thank you for adding the new permutation analysis (Fig. 6). I have just one minor comment: in line 183, I think the word "phenotypes" is missing: "behavioural *phenotypes* relevant to substance use disorder".

(Remarks on code availability)

I briefly skimmed the Github.

We thank both reviewers for their careful and expert review of our manuscript. We have fully addressed their comments, which has improved our manuscript.

When the two reviewers commented on a same issue, we **highlighted in red** our response to Reviewer #1 and pointed to this response when replying to Reviewer #2.

Reviewer #1 (Remarks to the Author)

Review of Tonnele et al 2025

The manuscript by Tonnele and colleagues investigates the genetic architecture of microbiome composition in a very large cohort of heterogenous rats. The authors identify several genotype-microbiota associations that are investigated at the locus level with functional candidate genes names. Additionally there is considerable analysis of cage, maternal and “indirect” genetic effects. All of these are important aspects of the study. Below are some suggestions and questions in regards to the analysis and conclusions

Microbiome questions

- What was the distribution of zero inflated data in each cohort?

Short answer: as stated in the main text, only the microbiome phenotypes that had a prevalence greater than 50% were considered for genetic analysis. Fig. 3 and SFig. 5 were wrong in our original submission, due to a bug in the code we wrote to plot these figures. We have now corrected both figures and you can see in those figures the prevalence (1 –proportion of zeroes) of the microbiome phenotypes. We thank both reviewers for noticing this issue.

More detailed answer: The code used to generate Fig. 3 and SFig. 5 in our original submission was wrong as it considered all microbiome phenotypes (prevalence above and below 50%). Since heritability was always calculated only for those phenotypes with a prevalence greater than 50%, R was recycling the calculated heritability values to give a heritability value to all the phenotypes. We have corrected the code, as can be seen on GitHub.

```
165/Figures/3b.prev_abund_herit.R
30 31 study(cue = all(study))
57 60
58 - # - Amelie comments -
59 61 prevs = get(paste("prevs", study, sep='_'))
60 62     match = match(paste(names(prevs), study, sep='_'), all_VCs_full$trait1)
61 63     cols = all_VCs_full[na.exclude(match), 'color_Ad1']
62 - #all_cols = c(all_cols, cols) #previously used to have all cohorts in one plot
63 - #all_prevs = c(all_prevs, prevs[!is.na(match)]) #previously used to have all cohorts in one plot
64 + prevs = prevs[!is.na(match)]
64 65     meds = get(paste("meds", study, sep='_'))
65 - #all_meds = c(all_meds, meds[!is.na(match)]) #previously used to have all cohorts in one plot
66 - #     plot(prevs, meds, col= cols) #previously used to have all cohorts in one plot
67 - # - end Amelie comments -
66 + meds = meds[!is.na(match)]
68 67
69 - # Plot
68 + # Plot
70 69     plot(prevs, meds, col= cols,
71 70         xlab = paste0("Prevalence in ", studytitle, " cohort"), ylab = "",
72 71         las = 1, cex.lab = 1.4, cex.axis = 1.25,
```

How were 0 data accounted for?

0 values were offset to 0.00001 prior to CLR transformation. While we recognize that there are more principled ways of handling zeroes, our replacement of 0s by arbitrary, small positive values is similar to “using the minimum proportional abundance detected for each taxon for the imputation of zeros” (Valles-Colomer et al. Nature 2021 <https://doi.org/10.1038/s41586-022-05620-1>), “zeros in the tables were adjusted by adding half of the lowest value in the table to each cell” (Gacesa et al. Nature 2022 <https://doi.org/10.1038/s41586-022-04567-7>), “Zeroes in the MetaCyc and in the GBM profiles were imputed by pseudo-count of one” (Manghi et al. Nature Communications 2024 <https://doi.org/10.1038/s41467-024-53934-7>). More importantly, we took several steps towards limiting and accounting for potential issues arising from zeroes:

- first and foremost, we reduced the number of zeroes by considering for genetic analysis only those microbiome phenotypes with a prevalence greater than 50%
- prior to filtering microbiome phenotypes based on prevalence, we filtered samples based on library size, excluding samples with low library sizes (i.e. many zeroes)
- we also fitted a fixed effect for library size in our models, to avoid any risk of zeroes reintroducing an effect of library size in lower prevalence taxa (Te Beest et al., 2021 <https://doi.org/10.1111/1755-0998.13391>)
- we used the conservative approach of breaking ties (when two or more rats had a zero value) at random during rank normalization. Breaking ties at random is expected to increase noise, which is expected to result in fewer heritable microbiome phenotypes and fewer significant microbiome-associated loci. Nevertheless, we found a large proportion of microbiome phenotypes to be significantly heritable and identified three replicated microbiome-associated loci.

Use of high 0 inflation with rank normalization could be problematic in a linear model. This is most evident in supplementary figure 5 and figure 3. There is not a relationship between relative abundance and H2. There are heritable microbiome (ASV?) that are low abundance that also have low prevalence. The latter indicate relatively rare ASV are included in the results. Figure 3 indicate that most of the ASV are low Heritability and low prevalence and low abundance. For the ASV highlighted as genetically regulated, what is prevalence and relative abundance?

We apologise for presenting incorrect figures in our initial submission. As explained above, we have now corrected them, and you can see that there are no microbiome phenotypes (ASVs or taxa) with a prevalence lower than 50%.

While the most heritable microbiome phenotypes are not the most abundant ones and the correlation between abundance and heritability was not significant, there was nevertheless a significant difference in terms of abundance between heritable and non-heritable phenotypes (one-sided t-test p-value = 0.001935 using the whole sample). We have now added this result to the main text.

- What was the total number of specific sequences identified at a phylogenetic level (i.e. family) for the whole study? Did this vary between sites?

The total number of microbiome phenotypes (ASV + species-level taxa + genus-level taxa + ... + class-level taxa) with prevalence greater than 50% was 569 in the NY cohort, 589 in the MI cohort, 523 in the TN1 cohort, 613 in the TN2 cohort (first paragraph of Results section 2 “Polygenic host genetic effects and their consistency across environments”) and 546 when all cohorts were analysed jointly (last paragraph of that section). About half of the microbiome phenotypes are ASVs and the other half are taxa.

- This is a large study. How were samples randomized into sequencing runs?

Sample randomisation was carried out by mixing a few hundred of the samples available at the time of processing (see below) in a very large polystyrene box, returning the samples to sample boxes and processing the sample boxes in a random order.

As shown in the table below, two batches of samples were processed in 2017 (one batch including randomised samples collected in 2015 and 2016, the other batch including randomised samples collected in 2015, 2016 and 2017) and the remaining seven batches were sequenced in 2019 (including randomised samples collected 2015-2018).

Qiita prep ID:	prep_4075	prep_4078	prep_6462	prep_6826	prep_6827	prep_6935	prep_7783	prep_7808	prep_7831
	year 2017	year 2017	year 2019	year 2019	year 2019	year 2019	year 2019	year 2019	year 2019
sample collected 2015	141	101	0	7	0	0	27	29	25
sample collected 2016	458	253	6	64	34	34	33	44	23
sample collected 2017	0	160	99	206	267	96	58	56	55
sample collected 2018	0	0	45	140	168	118	52	39	81
unknown collection date	13	5	18	77	59	22	362	362	340

We have now added this information to the Methods sub-section “Sample randomization, DNA extraction, library preparation and sequencing of the 16S amplicon”.

Rat cohort questions

- The protocols at each site were different. In supplemental table 1 details of different protocols are listed. The authors should be more transparent about the variation in diet, drug exposure, housing conditions and phenotyping which all could affect results. Do the results indicate that the loci identified are invariant to environmental changes?

This variation is suggested in the abstract and explicitly announced in the last paragraph of the introduction (“The four cohorts differed in terms of rearing facilities, age and fasting state when the cecal (gut) microbiome was sampled, among other known and presumably unknown differences (Fig. 1B and Suppl. Table 1)”). The issue is also presented in Fig. 1, whose caption points to STable 1 and Methods for extensive additional information. Thus, we feel we have already been very transparent about this issue.

Indeed, one of our important results is that the three replicated associations (Fig. 4) are conserved despite the environmental differences between the four cohorts. This is true, to a large extent, for polygenic effects too, as shown in Fig. 3. In our Discussion we wrote that “Our study is the first to examine the degree of consistency of polygenic host genetic effects across environments. We found consistent host genetic effects across cohorts. This result is consistent with a previous cross-cohort analysis of the effect sizes of GWAS- significant loci in humans and suggests that statistical power is the main limitation for replicating microbiome-associated loci in humans as in other species.” Despite the different experimental exposures in our study, we suspect that environmental conditions are still more uniform than in a study of humans, where environmental exposures are even more variable.

- The indirect genetic analysis is intriguing. The figure indicates a “genotype” effect but I interpret this to a “rat strain” effect as these are complex 8 way cross. The

genotype effect would be more appropriate for a specific test such as a knockout animal.

While genotype is relatively commonly used to refer to the genetic background of an outbred rodent, we agree that this term may seem odd for some readers. We have therefore changed “genotype” to “genetic background” in Fig. 6A.

“Strain” would not be correct, as it should be used for inbred animals only.

The model and results indicate that the effects of horizontal transmission (i.e. a within cage effect) could be dependent on genotype. It is not clear how robust these measures are and if these are random within cage effects.

While Mic-IGE are indeed a within-cage process, Mic-IGE and cage effects (i.e. environmental effects affecting all the rats in a cage in the same way) are not at all confounded in the NY and TN2 cohorts and not fully confounded in the MI and TN1 cohorts. We explain why in the extended and improved Methods section entitled “Experimental design and how it relates to confounding between genetic effects, cage and maternal effects”. The same section now includes a list of figures demonstrating that fitting Mic-IGE and cage effects jointly in a variance components model yields unbiased estimates of Mic-IGE:

1) **SFig. 6 and SFig. 7 in Baud et al. PLOS Genetics 2017**, based on a dataset where relatives were co-housed (greatest risk of confounding)

2) In the current study: **SFig. 12** shows that Mic-IGE p-values follow the expected uniform distribution for simulations in which DGE, cage and maternal effects but no Mic-IGE are simulated.

3) Following a suggestion from Reviewer 2, we also carried out a new permutation analysis, whereby we permuted 1,000 times the cage mates' assignments while keeping the cage assignments unpermuted (to still capture shared environmental effects, see corresponding Methods section). We analysed these permutations with the model including Mic-DGE, Mic-IGE, cage and maternal effects. Our results, presented in **Fig. 6C** (total genetic variance) and **SFig. 13A** (Mic-IGE), show that the total genetic variance and Mic-IGE estimates from these permutations are smaller than the same estimates from the unpermuted data either in all the permutations or in 999 out of 1,000 permutations, strongly arguing that our observed results are real. Altogether, all our results strongly support the reliability of our results on Mic-IGE even in the presence of cage effects.

• Additional question on maternal effects. Did the authors quantitate the maternal microbiome?

It is difficult to understand the seeding or transmission of the microbiota without an actual measurement.

Studying the transmission of the maternal microbiome would have been interesting but is outside the scope of the present study, hence we did not profile the maternal microbiome (except for the TN2 cohort, but the mothers are used not as mothers but as any other rat).

Maternal effects in our study were only modelled because they were expected to contribute to microbiome variation and because they are partly confounded with DGE. They are modelled as random effects based on the identity of the mothers, not based on the mothers' microbiomes.

The model described in the methods uses W' to model maternal effects. It isn't clear how many offspring per dam were included in the analysis. Are dams bred multiple

times? Given the size of the study, it could be assumed “yes” but clarity is appreciated. If bred multiple times did the sires change between breeding’s? Did littermates get sent to different locations?

The vast majority of mothers having produced rats for the MI, NY and TN1 cohorts (74%, 45% and 80% respectively) had 6 offspring: one male and one female that went to the MI cohort, one male and one female that went to the NY cohort, and one male and one female that went to the TN1 cohort. The male and female offsprings were never housed together.

Another small fraction of mothers (12%, 25%, 0%) had 4 offsprings only, spread across cohorts and cages similarly to what is described above. Finally, the remaining mothers had one, three, or in very rare cases 8 offsprings (different litters from the same father), also spread across cohorts and cages.

On the other hand, the mothers of the TN2 cohort (which were in the TN1 cohort, see Fig. 1), mostly (72%) had 4 offsprings, the two males being usually housed together and two females housed together.

This information is now summarised in the Methods section “Experimental design and how it relates to confounding between genetic, cage and maternal effects”:

How are half sibs and siblings from different generations dealt with in the analysis of direct and indirect genetic effects?

To address this comment, we have extended and improved the Methods section “Experimental design and it relates to confounding between genetic, cage and maternal effects”, including a list of the figures that support the accuracy of our genetic (DGE and IGE) estimates even in the presence of relatedness and maternal effects (SFig. 12, Fig. 6C and SFig. 13, Fig. 6D and SFig. 14).

For the GWAS of host genetic effects, to account for confounding by the other causal loci in the genome, we included a random term for polygenic/background DGE, which is the state-of-the-art method used in human and animal genetics. This quantitative approach not only addresses issues like sibs and half sibs but also the more distant relationships (various types of cousins) that are expected when rats are derived from a breeding colony with a few dozen breeding pairs.

• The polygenic host genetic effects also include a cage effect term (which should be confounded with maternal effects?)

To clarify, random terms for polygenic host genetic effects (Mic-DGE), cage and maternal effects were always included in our analyses. We have now expanded and improved the Methods section “Experimental design and how it relates to confounding between genetic, cage and maternal effects”.

Analysis

• Locus zoom (previously SFig. 10, now SFig. 11) indicate a genotyping gap on chr 10 at 102-103 Mb. Is there a technical reason for the gap?

Essentially no, this is not a technical issue but rather a result of the region being fixed in the outbred population.

- Many variants are detected, in the region, between the eight inbred founders of the HS population. Almost all of them distinguish the WN/N founder from the seven other founders, which are IBD in the region.

- At these WN/N specific variants, all the outbred rats are homozygote for the reference (non-WN/N) allele, showing that the WN/N haplotype was lost locally during the generations of outbreeding.

- There are a handful of variants sparsely distributed in the region that are unique to one of the seven (non-WN/N) founders and that can be seen in the raw (unimputed) genotype data but not in the imputed data, due to the imputation process. We have now added this information to the caption of SFig. 11, which shows a zoom-in on the locus.

• The simulations are difficult to understand. In supplemental figure 14 the authors state- “The reason why (A) and (B) are different is most likely that, in the NY cohort, the rats were housed 2 to a cage, hence had no shared cage-mate.” However, the figure panels look very similar. Can they clarify specific differences?

To address this comment, we have taken three actions:

1) We have simplified SFig. 14 to only show DGE estimates and, thus, match Fig. 6D (SFig. 14 is for the MI cohort, Fig. 6D for the NY cohort). In addition, we have further simplified the two figures to focus on the results from simulations in which the correlation between DGE and IGE was positive (we studied the extremes: 0 and 1). The focus on positive correlations is based on the hypothesis that microbiome transmission mediates the majority of Mic-IGE, as we explained in the main text. Fig. 6D (NY cohort) and SFig. 14 (MI cohort) are now qualitatively and quantitatively essentially the same.

2) We have added a section in the Methods describing these simulations; this was previously missing.

3) We have added, in the main text, a few sentences explaining the intuition behind why DGE are underestimated.

4) For the 26 microbiome phenotypes significantly affected by Mic-IGE (FDR < 0.1), we have also quantified how much bigger Mic-DGE were when they were estimated from a model accounting for Mic-IGE, compared to a model not accounting for Mic-IGE.

• The abstract focuses on ST6GAL1 and immunity. Do the authors have any data from these animals to support this statement—“this suggests ST6GAL1’s effects on IgA nephropathy and COVID-19 breakthrough infections may be mediated by *Paraprevotella*.” The genotype- microbiota association is strong but has not been tested in this paper. If data is not available, please remove from the abstract as this is not directly supported by the data in the current manuscript.

We have now deleted the sentence “Given *Paraprevotella*’s known immunity-potentiating functions, this suggests ST6GAL1’s effects on IgA nephropathy and COVID-19 breakthrough infections may be mediated by *Paraprevotella*.” We included the text “both of which have been linked with immune and infectious diseases” in the updated abstract, to convey the possible implications of our results. If the reviewers feel this is still too much, we will remove “both of which have been linked with immune and infectious diseases”.

Reviewer #2 (Remarks to the Author)

This manuscript by Tonnele, Chen, and colleagues analyses host genetic influences

on the gut microbiome in several cohorts of genetically diverse rats. This work addresses the open question of just how much host genetics influence the gut microbiome. One end of the spectrum is that microbiome heritability is nearly universal, but we need very large sample sizes and longitudinal data to detect this heritability (Grieneisen et al. 2021). The other end of the spectrum is that microbiome heritability is generally quite modest (Rothschild et al. 2018, Gacesa et al. 2022).

This paper provides support for appreciable microbiome heritability. Finding microbiome-associated loci that are consistent across cohorts is a compelling argument for a real genetic signal. Relating the hit involving *Paraprevotella* to human data was a nice connection.

Overall, the analysis was thoughtful, the methods were clear, and in general, the claims were supported by the data presented.

I admit that I remain sceptical about the final section relating to indirect genetic effects, namely that accounting for IGEs can increase the amount of explained genetic variance so dramatically (Fig. 6C). One idea for increasing confidence in this analysis is to repeat it using scrambled cage assignments. If IGEs fail to explain any additional variance when the model uses incorrect cage labels, I would find it reassuring that the methodology is not artificially inflating the amount of explained genetic variance. I believe this is different from the simulations shown in Fig. 6D. This analysis is not mission-critical.

We thank the reviewer for this excellent suggestion, which we feel really strengthened our study. As detailed in the response to Reviewer #1 above, we have now carried out the suggested analysis, whose results (Fig. 6C and SFig. 13A) further support the impact of Mic-IGE on microbiome phenotypes.

One question that remains for me (and does not need to be addressed by the authors, though it would be interesting to hear their thoughts) is how to reconcile microbiome heritability with how dynamic we know the microbiome can be. In other words, we know that the microbiome changes day-to-day and week-to-week in response to factors like diet, medications, whom we live with, etc. And yet heritability implies stability, i.e. (fixed) host genetics has some stable effect on the microbiome (although one could imagine temporally dynamic genetic influences). This question could be explored with longitudinal data but unfortunately not in this dataset. While we agree that host genetics (i.e. genotypes) is fixed at birth, we do not agree that host genetic effects are fixed: their magnitude, when it is expressed as a function of the total phenotypic variance (e.g. heritability for polygenic host genetic effects), depends on the amount of environmental variance. So, for example, while rats in our study cycled through day and night (i.e. time of day was a dynamic environmental factor), they were all sacrificed at roughly the same time of day, which greatly limited the amount of variation in microbiome phenotypes due to the circadian rhythm and maximised host genetic effects. In another study where samples might be collected at different times of day, the careful recording of the time of sampling would permit modelling the effect of time of day on the microbiome and, accounting for it as a fixed effect and not considering the corresponding variance in the calculation of heritability, as is common practise, time of day would not contribute to phenotypic variance and heritability would be maximised.

Longitudinal data permit to distinguish between permanent environmental effects and genetic effects. In addition, in settings where the environment is dynamic (e.g. in the wild), permit exploring the interplay (interaction) between genetics and environment. This was indeed not possible with our dataset.

We have not added this full discussion to the Discussion section of our manuscript to keep it short, and because none of our data directly address these issues, but we would be happy to do so if the reviewers thought it was important. For now, we have only included the following text: “The large sample size we used (N = 3,784) and the limited environmental (experimental) variation in our study explain why we detected so many heritable microbiome phenotypes. More specifically, some relevant factors did not vary at all (e.g. diet) while others did vary but did not contribute to the phenotypic variance (e.g. time of day; all samples were collected at roughly the same time of day).”

I have no major concerns and commend the authors for a solid manuscript. Please see below for minor concerns and additional comments.

Minor Concerns

- Line 60: “Given *Paraprevotella*’s known...” Interesting, but perhaps too speculative for the abstract (see comment about line 497 below).

We have now deleted this sentence. Please see our response to the same comment from Reviewer #1 above.

- Lines 80-82: “Only two microbiome-associated...” It would be nice to include citations for this sentence.

We have included 3 references for LCT and 2 for ABO.

- Line 183: “behavioural and physiological phenotypes”. Like what? It would be helpful to list a few examples of these phenotypes.

We have added examples: delay discounting, cue and context conditioning, nicotine self-administration for behavioural phenotypes adiposity, kidney and liver weight, glycemia for physiological phenotypes.

- Fig. 3B: why are there dots with prevalence below 0.5? I thought only ASVs with prevalence > 0.5 were tested (lines 173-174). Also, this plot is okay - but wouldn't it be more straightforward to just plot median abundance versus heritability?

As detailed in our response to the same comment by Reviewer 1's, there was an error in the code we used to generate Fig. 3B and SFig. 5, which we have now corrected. This reviewer is correct that only microbiome phenotypes with prevalence greater than 0.5 were considered for genetic analysis.

The figure permits to visualise the relationship between prevalence and heritability and the relationship between abundance and heritability, which can both be thought to be relevant *a priori*.

- Fig. 3D: I think this is a very important panel. My interpretation is that, in general, there is little concordance in heritability across cohorts (even though the authors point out that certain pairwise correlations are significant). Alternatively, one could argue that the global concordance may not be great, but there is concordance for the hits discussed in Fig. 4. Could the authors label these hits (from Fig. 4) in this panel?

Our interpretation is also that, in general, there is little concordance. We have edited the main text to more clearly convey this interpretation.

Whether there is more concordance for the microbiome phenotypes for which we detected significant associations is not what we meant to investigate and, in fact, we expected only a little bit more concordance for those phenotypes, because significant associations typically explain only a fraction of the heritable variance (Baud et al. Nature Genetics 2013, also in HS rats).

Nevertheless, we have coloured the dots corresponding to the most significantly associated ASV/taxon at each of the three replicated associations using the same colouring scheme as in Fig. 4 (ASV_3613 in purple, ASV_18566 in blue, and ASV_5163 in orange). More specifically, the dots corresponding to these ASVs/taxa were coloured only when there was a genome-wide significant association in both cohorts of the cohort pair considered (NY, MI and TN2 for ASV_3613; NY, MI and TN1 for ASV_18566; NY and TN1 for ASV_5163). The updated Fig. 3D shows that heritability is relatively concordant for several of these cases, but not all.

- Line 260: “This analysis did not...” It might be nice to include a figure panel showing this point.

This is now in SFig. 6.

- Fig. 5D-F: I don't follow. If the abundance of these ASVs is associated with *St6galnac1*, don't we expect to see higher ASV abundance with duplications or triplications? At best, this is true only in panel F. Also, what is the difference between “single copy” and “het”?

“Single copy” refers to the genotype of rats that have a single copy of the *St6galnac1* gene. “Partial dupl/tripl”, which we have now changed to “Triplication”, refers to the genotype of rats who have two additional full-length copies of *St6galnac1* in their genome (variant shown in Fig. 5B and 5C, inherited from the WN/N founder). Since there are very few rats that are homozygote for the triplication, the association between the *St6galnac1* copy number variant and *Paraprevotella*, *Akkermansia muciniphila* and *Muribaculum intestinale* is based on the comparison between the single copy homozygotes and the heterozygotes.

Whether the abundance of a bacteria is expected to increase or decrease with additional gene copies depends on how the gene products (here enzymatic activity) affect the bacteria. To address this reviewer's comment, we have extended our discussion of the most likely effect of *St6galnac1* on *Paraprevotella*, *Akkermansia muciniphila* and *Muribaculum intestinale*.

- Lines 432-434: “We demonstrated that host genetic effects...” I agree that the 3 loci look real, but the heritability scatterplots (Fig. 3D) argue against the first half of the sentence. I know the authors also use genetic correlation to argue for concordant genetic effects, but it's hard for me to understand high genetic correlations without concordant heritability estimates.

We now specify, in the Results section, that “consistent” refers to high genetic correlations, and have also added two sentences to explain why a low concordance of the heritability values is not inconsistent with high genetic correlations.

- Lines 484-486: “Together with evidence...” The argument is that *Paraprevotella*

makes IgAN worse. Any thoughts about why IgAN patients have more *Paraprevotella* in the first place? Because of mutations in *ST6GAL1*?

Indeed, this is supported by the association we found between *ST6GAL1* and *Paraprevotella* in humans.

We have tried to clarify this by editing our Discussion: “*A. muciniphila* is capable of desialylating and more generally deglycosylating IgA, which induces IgAN. Our results showed that variants in *ST6GAL1* affected the abundance of *Paraprevotella* in the human gut and they suggested *Paraprevotella* has similar desialylating and deglycosylating activities as *A. muciniphila* in the rat gut. We therefore suggest that *Paraprevotella* can desialylate and deglycosylate IgA and induce IgAN.”

- Line 497: “Thus, our results support...” Interesting but highly speculative. Seems like this claim is supported by evidence external to this study, namely that *Paraprevotella* degrades trypsin (leading to decreased viral entry into epithelial cells) and that *Paraprevotella* may improve vaccine efficiency. It feels like this summary sentence is just a little too far afield from the findings actually presented in the paper, but maybe it’s okay since it’s in the discussion.

We agree. We have now phrased this as something more hypothetical in our revised manuscript (“might mediate”, “if *Paraprevotella* [...] influenced”) and deleted “Thus, our results support”.

- Line 687: “each microbiome phenotype was quantile (i.e. rank) normalised, with ties were broken at random.” This could be a problem when there are many zeros. Were they many zeros?

The prevalence of the microbiome phenotypes considered in genetic analyses was always greater than 50%.

As explained in response to a comment by Reviewer #1, breaking ties is a conservative approach that did not prevent use from finding that a large proportion of microbiome phenotypes were heritable and from detecting three replicated microbiome-associated loci.

Praiseworthy

- Lines 172-173: “We analysed the different cohorts separately...” Analysing the cohorts separately (before analysing them jointly) was nice. Cross-cohort consistency is a compelling way to increase one’s confidence in the hits.

- Lines 179-182: “Hence, the heritability of microbiome...” Definitely agree! This was a good analysis.

- Lines 184-185: “ASVs tended to have higher heritability...” Nice result.

- Line 339-342: “In a human cohort...” Nice connection to human data.(Remarks on code availability)

Thank you for pointing these out!

I took a look through the Github for reproducing figures. It seemed reasonably thorough and well-organized. I did not try running any code.